# MIR-Bench: Can Your LLM Recognize Complicated Patterns via Many-Shot In-Context Reasoning?

**Kai Yan,** * **Zhan Ling, Kang Liu, Yifan Yang,**
**Ting-Han Fan, Lingfeng Shen, Zhengyin Du, Jiecao Chen**
ByteDance Seed
kaiyan3@illinois.edu

## Abstract

The ability to recognize patterns from examples and apply them to new ones is a primal ability for general intelligence, and is widely studied by psychology and AI researchers. Many benchmarks have been proposed to measure such ability for Large Language Models (LLMs); however, they focus on few-shot (usually <10) setting and lack evaluation for aggregating many pieces of information from long contexts. On the other hand, the ever-growing context length of LLMs have brought forth the novel paradigm of many-shot In-Context Learning (ICL), which addresses new tasks with hundreds to thousands of examples without expensive and inefficient fine-tuning. However, many-shot evaluations often focus on classification, and popular long-context LLM tasks such as Needle-In-A-Haystack (NIAH) seldom require complicated intelligence for integrating many pieces of information. To fix the issues from both worlds, we propose MIR-Bench, the first many-shot in-context reasoning benchmark for pattern recognition that asks LLM to predict output via input-output examples from underlying functions with diverse data format. Based on MIR-Bench, we study many novel problems for many-shot in-context reasoning, and acquired many insightful findings including scaling effect, robustness, inductive vs. transductive reasoning, retrieval Augmented Generation (RAG), coding for inductive reasoning, cross-domain generalizability, etc. Our dataset is available at https://huggingface.co/datasets/kaiyan289/MIR-Bench.

## 1 Introduction

The tremendous success of Large Language Models (LLMs) in recent years [60, 27, 28] has brought the prospect of human-level Artificial General Intelligence (AGI) into sharper focus [28]. With such success, researchers have shifted their focus from syntax- and word-level traditional Natural Language Processing (NLP) tasks such as named entity recognition [56, 42], sentiment classification [74, 79] and translation [50, 76] onto abilities once considered unique to humans, such as **the ability to recognize patterns and apply them to new examples across diverse contexts** (instead of only in predefined domains such as those mentioned above). Such ability, including *inductive reasoning* [23] (explicit recognition of abstract rules) and *transductive reasoning* [63] (implicit recognition from local examples), measures the generalization power of an intelligence [11] and are considered as very important mental abilities [33]. Thus, they are long studied by the cognitive science community [7, 24], adopted in IQ tests for human [18], and is recently used as a measurement for the state-of-the-art LLMs such as o1 [28] and o3 [59] to show their level of intelligence. Such abilities are also vital for future LLM generalist agents [67], where the agents must perceive and summarize the inherent logic of the environment and act according to past successful experiences.

---

*Corresponding author; work done during internship at ByteDance Seed.

39th Conference on Neural Information Processing Systems (NeurIPS 2025) Track on Datasets and Benchmarks.

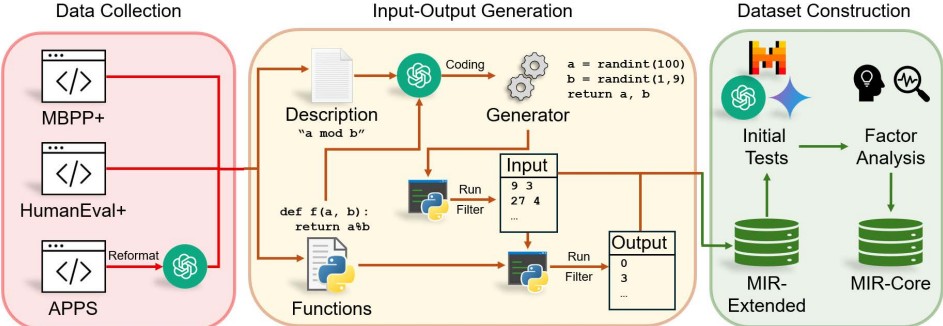

Figure 1: A high-level illustration of our data generation pipeline. We first collect functions from existing coding benchmarks, then let GPT-4o-0806 write data generator for each function; we then run the data generator to produce input shots, and combine them with ground truth function to produce output shots. With input and output shots, we concatenate them and build MIR-extended; then, with initial tests on several models, we study the factor for what makes a pattern recognition problem benefit from many-shot, and build MIR-core based on selection with the factors.

While many pattern recognition benchmarks [5, 51, 43] for LLMs have been proposed, such as ARC variants [11, 32, 88] and inductive reasoning benchmarks such as WILT [5] and DEER [91], they all focused on few-shot In-Context Learning (ICL) with typically <10 examples. While pattern recognition from fewer examples may imply stronger reasoning ability, some underlying rules in real-world problems are inherently too complicated or ambiguous for a few examples. For instance, consider a quadratic curve with clipping. With three examples, it is unknown whether the curve is sampled from a circle or a quadratic curve, let alone a clipped one; however, with 300 examples, not only the quadratic function is clear, but the special clipping rule are also very likely to be retrieved. LLM should handle such long-context, many-example cases as well as few-shot reasoning.

In fact, the scaling of the amount of ICL data is in line with the trend of the LLM community striving to expand the context length [62, 77] for super-human problem-solving efficiency. It is with this trend that a new paradigm emerged recently: Many-Shot ICL, which typically uses hundreds to thousands of examples for test-time task learning without using expensive and relatively data-inefficient fine-tuning [1]. However, many-shot evaluations are mostly focused on classifications [45, 6, 99, 46, 101], which is a very limited area of pattern recognition[2]. Other standard long-context LLM tasks, such as needle-in-a-haystack (NIAH) [31], are more of a retrieval problem than gathering understanding from many pieces of clues. With all these blanks in LLM evaluation (see Tab. 1 for a comparison with the most related benchmarks, and Tab. 3 in Appendix B for a more complete version), we must ask: *How to evaluate the LLM's ability to aggregate many pieces of information from many examples to perform pattern recognition on various complicated problems?*

To address the problem above and fix the limitation of existing LLM evaluation from both pattern recognition and the many-shot/long-context community, we propose MIR-Bench, a large and diverse **M**any-shot **I**n-Context **R**easoning benchmark, where LLMs are given examples of input-output examples generated by an underlying unknown function with diverse input-output forms, and need to recognize the patterns for predicting the output for new input.

The benchmark is generated by the following pipeline as illustrated in Fig. 1: 1) we collect functions from introductory-level coding benchmarks including HumanEval+ [49], MBPP+ [49] and APPS [25]; 2) we use GPT-4o-0806 to write code as data generators that produces input-output pairs, and execute them to generate ICL shots and test input; 3) run ground truth function with generated inputs for ground-truth outputs; 4) use scripts to build prompts for target problem, and filter out problems with too long shot length or insufficient input-output diversity. With such procedure, we propose two sets of problems: **MIR-Core** and **MIR-Extended**, which contains 3000 problems (300 functions × 10 test cases), and 6930 problems (693 functions × 10 test cases) respectively, and can be easily supplemented by generating more test cases. The former is selected from the latter and contains the problems that LLM benefits the most from many-shot (see Sec. 4.2 for details).

---

[2]Mostly transductive reasoning in these works.

Table 1: The topic, validity and reproducibility comparison between our work and prior benchmarks. See Tab. 3 in Appendix B for a complete comparison. To save space, we abbreviate "Many Shot" as MS, "Pattern Recognition" as PR ($\triangle$ for "classification only"), "Prob." as problems, and "I/O Div." as "Input/Output Diversity" ($\geq 2$ **different input-output types, e.g., given an array and output an integer, or given a pair of strings and output a choice**). "Gen." means "Generative", which means whether new test cases can be easily created without much human effort. "LB" means available leaderboard, and "EE" means "Easy Evaluation", i.e., whether a pipeline for evaluating any given new model exists. "New Data" means if the input-output data never appears in existing benchmarks and thus is secured against data contamination; benchmarks with "New Data" being $\times$ is a compilation of existing benchmarks. Note, the counting of #PR Problems and "Gen." take different target input-output for the same function into account, but **not different sets of shots.**

| Benchmarks | MS | PR | # PR Prob. | I/O Div. | Max # Shots | Gen. | LB | EE | New Data |
|---|---|---|---|---|---|---|---|---|---|
| HELMET [92] | ✓ | △ | 500 | ✓ | ∼10K | ✗ | ✗ | ✓ | ✗ |
| LongICLBench [46] | ✓ | △ | 3000 | ✗ | ∼2000 | ✗ | ✓ | ✓ | ✗ |
| ManyICLBench [101] | ✓ | △ | 1000 | ✓ | 7252 | ✗ | ✗ | ✓ | ✗ |
| LMAct [67] | ✓ | ✓ | N/A * | ✗ | 256 | ✓ | ✗ | ✓ | ✓ |
| LongBench [4] | ✓ | △ | 400 | ✓ | 600 | ✗ | ✓ | ✓ | ✓ |
| KORBench [51] | ✗ | ✓ | 50 | ✓ | 3 | ✗ | ✓ | ✓ | ✓ |
| ARC [11] | ✗ | ✓ | 800 | ✗ | 3 | ✗ | ✓ | ✓ | ✓ |
| WILT [5] | ✗ | ✓ | 50 | ✗ | 30 | ✗ | ✓ | ✓ | ✓ |
| LogicVista [87] | ✗ | ✓ | 107 | ✓ | 10 | ✗ | ✗ | ✓ | ✓ |
| MIRAGE [43] | ✗ | ✓ | 2000 | ✓ | 8 | ✓ | ✗ | ✗ | ✓ |
| **MIR-Bench (Ours)** | ✓ | ✓ | 6930 | ✓ | 2048 | ✓ | ✓ | ✓ | ✓ |

* LMAct has only a few tasks, but it is interactive and thus hard to count the number of problems.

To evaluate the long-context intelligence level of different LLMs, we test a variety of the cutting-edge LLMs on our benchmark, and find our benchmark to be both challenging and discriminative, as model performance vary greatly, but none saturates on our benchmark. We have also conducted the following important and extensive empirical studies with our benchmark:

1. What are the factors for ICL performance change with the number of shots (Sec. 4.2, 4.3)?
2. Is LLM a better inductive reasoner or transductive reasoner? (Sec. 4.4)
3. How robust is LLM's pattern recognition ability against errorneous examples (Sec. C.1)?
4. Does the inductive paradigm of first coding, then executing code for results [10] work for many-shot in-context reasoning (Sec. C.2)?
5. Does Retrieval-Augmented Generation (RAG) [19] help many-shot reasoning (Sec. C.3)?
6. Can LLMs generalize from "meta-shots", i.e., other inductive reasoning examples (Sec. C.4)?

In conclusion, our key contributions are: 1) We propose MIR-Bench, the first large-scale, diverse, non-compilation many-shot pattern recognition reasoning benchmark, which fills in the blank for both many-shot and inductive / transductive community; 2) We build a novel automatic pipeline for generating new tasks from existing coding benchmarks without using existing corpus as input/output (i.e., no data leakage issues); 3) We perform empirical study on many important problems overlooked by previous works and gained important insights on LLM's many-shot / long-context intelligence.

## 2 Related Work

**Long context LLMs.** Recent remarkable success of LLMs have given rise to expectations for LLMs to complete more difficult tasks, such as summarization of a whole book [8], modification over a complex code repository [30], test-time improvement [100] and journey learning [64]. To make sufficient room for related context and meet such demands, researchers have scaled up LLM models and data [95, 61], and proposed novel encoding methods such as Rotational Position Embedding (RoPE) [77], YaRN [62] and LongRoPE [15]. With such designs, LLMs have entered the long-context era where the LLM context lengths can reach 128K [27, 90, 17], 2M [80], or even an infinite number of tokens [57], enabling the novel many-shot ICL [1] paradigm. To evaluate such models, many

benchmarks have been proposed to evaluate LLM's long-context ability [83, 44, 97], such as Question-Answering [69, 44], coding [97, 16], math [3, 97], retrieval [31, 26, 83] and summarization [69, 3]. However, very few long-context benchmarks consider inductive/transductive reasoning tasks. Among them, LongBench [4] only contains two many-shot classification tasks and few-shot summarization / QA tasks with existing dataset, while BABILong [36] only considers simple inductive reasoning from a few examples scattering in the long context. In contrast, our benchmark is a more diverse and large-scale evaluation for long-context, many-shot inductive/transductive reasoning.

**Many-Shot In-Context Learning (ICL).** Many-Shot ICL [1] is an emerging ICL paradigm where LLMs learn to complete new tasks with hundreds to thousands of examples (instead of the usual $< 10$ examples [43, 11, 88]) given in its context. Compared to Supervised Fine-Tuning (SFT), many-shot ICL makes full use of the current models' long-context capability, is much more flexible with higher computational and data efficiency [1], and is inherently immune to catastrophic forgetting [35]. There are a large number of many-shot ICL empirical studies [6, 99, 75, 98] with several benchmarks [92, 46, 101, 67] containing many-shot ICL tasks; however, most of them only focused on classification [99, 46, 45, 29, 6], a very limited type of problems. While there are several works that studies decision-making [67], math [1], instruction following [98] and LLM judges [75], none of the existing works has studied general inductive/transductive reasoning, the important measurement of intelligence level [11]. Also, most of the existing many-shot ICL evaluations are not diverse enough, which means they only have one pair of input-output types [67, 29]. Our work, on the contrary, measures LLM's intelligence level using pattern recognition with diverse input-output types.

**Inductive reasoning and transductive reasoning.** Inductive reasoning [23] is the ability to explicitly summarize general rules from examples, while transductive reasoning [63] implicitly generalizes from existing examples to new instances in a "K-nearest neighbor" manner. Both abilities feature *recognizing patterns from examples*, which are very important for humans (and future LLM generalist agents) to perceive the world via experiences [67]. Thus, both abilities have been widely studied as primal mental abilities of human intelligence [33, 34] in IQ tests [18] and cognitive science [7, 24] long before LLMs existed. As LLMs approach human-level intelligence recently, pattern recognition also becomes an important task in analyzing LLM's intelligence, especially for theoretical and empirical studies on ICL [20, 2, 82]. Thus, many inductive [65, 84] / transductive [48] reasoning-based approaches and benchmarks have been proposed [11, 51, 43, 87, 5]. The most representative one is the Abstract Reasoning Corpus (ARC) [11] and its variants [32, 88], which is recently used to demonstrate the intelligence level of OpenAI o3 models [59]. The most similar work to ours is FIND [68], which tests LLM's ability to induce and interpret underlying functions composed by atomic functions; the LLMs interactively probes input-output pairs during evaluation. However, almost none of them are designed for many-shot scenario (except for mini-SCAN [65, 37] dataset appearing in Qiu et al. [65]). By filling in this gap, our many-shot in-context reasoning benchmark not only enables the LLM inductive/transductive reasoning community to catch up with the long-context era, but also tests the ability of LLMs to gather information from thousands of pieces of data, much more than existing pattern recognition problems [11, 43].

# 3 MIR-Bench

In this section, we will introduce our MIR-Bench in details, with Sec. 3.1 discussing the formulation of the problems evaluated and Sec. 3.2 introducing the pipeline with which we build our benchmark.

## 3.1 Problem Formulation

The goal of the problems in our benchmark is for LLMs to predict the output for a new input given a list of examples. More specifically, assume we have an underlying function $y = f(x)$, where $x$ and $y$ can be arbitrary data. [3] Assume for $f$ we have a set of $n$ known example input-output pairs $(x_1, y_1), (x_2, y_2), \ldots, (x_n, y_n)$, and a new input $x_{new}$; then, the LLMs' input will be $[c_1, str(x_1), str(y_1), str(x_2), str(y_2), \ldots, str(x_n), str(y_n), c_2, x_{new}]$, where $[\cdot, \cdot, \ldots, \cdot]$ is a string concatenation, $c_1$ and $c_2$ are general context prompts (e.g. "You are an expert in reasoning", or "Here is the target input"; see Appendix F for details), and $str(\cdot)$ is the string representation plus an "Input: " prefix for $x$ and "Output: " prefix for $y$. LLMs can output arbitrary rationale; however, they must

---

[3]In our implementation, $x$ is a dictionary with key being string (variable names) and values being arbitrary Python list, tuple, dictionary or/and number, while $y$ can be any Python list, tuple, dictionary or/and number.

end their answer with $\text{str}(y_{\text{new}})$, where $y_{\text{new}} = f(x_{\text{new}})$. The answer is extracted with rule-based scripts, and exact match will be performed to determine the LLM's performance in accuracy. See Appendix F.6 for details on answer extraction.

## 3.2 Benchmark Construction

The construction of our benchmark can be decomposed into four steps: function collection, input generation, output generation, and prompt building.

**Function collection.** We begin by collecting introductory-level coding problems from three coding benchmarks: Humaneval+ [49], MBPP+ [49], and APPS [25]. We use the whole Humaneval+ and MBPP+ dataset (164 and 378 problems respectively); for APPS dataset, we select problems from its training dataset with difficulty level "introductory" (2640 problems). We choose the solution for coding tasks as the underlying patterns for input-output. This is because we intend to involve as little prior knowledge as possible and separately test the pattern recognition ability. While having diverse data source such as math and text-based logical reasoning problems can be beneficial for evaluating LLMs' real-world reasoning ability, it may also introduce unexpected involvement of LLM's other abilities (e.g. math). With such a source of data, the questions in our benchmark have highly diverse difficulty level and input-output modalities; see Appendix. E for dataset statistics.

Note, solution functions for introductory-level coding problems are not necessarily easy to induce. For example, consider the following problem: *Given an input string and let 'a'=1, 'b'=2, ..., 'z'=26. If we see strings as the product of alphabets, output the last digit of the result; e.g., $f(bab) = (2 \times 1 \times 2)$ mod $10 = 4$, $f(zc) = (26 \times 3)$ mod $10 = 8$*. The solution function $f$ is a one-liner:

```
def f(s):  return reduce(lambda x, y:  (x * (ord(y) - 96)) % 10, s, 1)
```

. However, it is highly non-trivial to induce with only input-output pairs. A non-introductory level problem, such as dynamic programming with multiple functions and arrays of input, could be almost impossible to guess with input-output pairs even for humans. In our experiments, we find that introductory-level problems are already sufficiently challenging.

We ensure that each solution code is a single function without wrapping solution class or test statement; for codes in APPS that do not conform to this standard, we ask GPT-4o-0806 to rewrite the code given problem input and the solution code (See Appendix F.3 for prompts).

**Input generation.** We use GPT-4o-0806 to automatically generate inputs for each function acquired in the last step, for which prior works [70, 47] usually directly generate input data. However, such method is not only non-scalable, but also prone to errors such as input format mismatch. To address this issue, we prompt GPT-4o-0806 to first generate "data generators" for each problem (See Appendix F.4 for prompts), then run each generator in Python interpreter for data. We generate 20000 shots and 10 test cases for each problem, which is impossible to acquire with prior methods. We wrote the prompt such that the test case is supposed to be slightly harder (e.g. with larger numbers / longer lists) than the shots. In this step, we filter out problems with the generated input too identical ($\leq 4096$ different shots out of 20000), duplicate test cases, or test cases appearing in the shots.

**Output generation.** With input generated, we write a script to stitch generated input and ground truth function $f$ in the same Python script, and run them in the intepreter to acquire ground-truth output. In this step, we filter out problems with floating number output, unless the precision is fixed across all shots by rounding, given by input, or unimportant for exact matching (e.g. the function is to output absolute value). We also filter out problems with too low output diversity ($\geq 50\%$ of the shots having the same answer), and problems with invalid output due to code error.

**Prompt building.** In this step, we use Python scripts to automatically stitch input-output pairs with task description to generate final input for LLMs. Finally, we also filter out problems that are unsolvable (either too difficult or data coverage are insufficient) for LLMs, which are the problems that have 0 accuracy for all five models {GPT-4o-0806, GPT-4o-mini-0718, Gemini-Pro 1.5-002, Gemini-Flash 1.5-002, Mistral-Large-2} across $\{4, 8, 16, 32, 64, 128, 256, 512, 1024, 2048\}$ shots in 10 test cases. We choose 2048 as the maximum number of shots as most LLMs reaches its context limit at this point (see Tab. 8 for details). After this step, we have 693 valid functions, each with 10 test cases; these problems are the content of our benchmark version MIR-Extended. Within this version, we select 300 problems that are challenging and can largely benefit from many-shot; See Sec. 4.2 for details.

# 4 Experiments

In this section, we will introduce general performance of existing models on our benchmark and a series of exploratory experiments which gives novel insights. We first introduce the main results on our MIR-Extended benchmark in Sec. 4.1; then, we explore factors that indicate whether a problem can benefit from many-shot, and build MIR-Core in Sec. 4.2. We further conduct more in-depth analysis on important properties of LLM's many-shot intelligence in several aspects on MIR-Core in Sec. 4.3 and Sec. 4.4. We defer more empirical ablation and analysis to Appendix C.1 to C.4.

## 4.1 MIR-Extended

**Evaluation setup.** We evaluate a set of 15 LLMs with context window $\geq$ 128K tokens on our MIR-Extended benchmark with 693 different function and 10 test cases per function (6930 problems in total). The evaluated LLMs are: {OpenAI-o1-preview-0912, OpenAI-o1-mini-0912, GPT-4o-0806, GPT-4o-mini-0708, Gemini-Pro 1.5-002, Gemini-Flash 1.5-002, Gemini-Flash 2.0, Claude-3.5-Sonnet, Claude-3.5-Haiku, Claude-3-Haiku, Claude-3-Sonnet, Qwen2-72B-Instruct, Mistral-Large-2, Moonshot-128K, GLM-4-Plus} by invoking official APIs; see Appendix F.1 for detailed prompts. We use greedy decoding (with temperature 0) for evaluation (See Appendix D.1 for ablations on the robustness of evaluation), and use exact match accuracy as the metric with rule-based extraction of the answer from LLM's response (See Appendix F.6). Each model is evaluated with $\{4, 8, 16, \dots, 2048\}$-shot with shots uniformly randomly sampled from 20000 shots generated in Sec. 3.2. Importantly, to avoid possible difficulty fluctuation among different number of shots due to sampling, we ensure *the examples in test cases with more shots are supersets of those with less shots* (except for errorneous shots in Appendix F.7). Thus, the information given in the input is strictly increasing with more shots.

**Results.** Fig. 2(a) illustrates the performance of all 15 LLMs on our MIR-Extended benchmark. The performance of the LLMs varies greatly; among all models, o1-mini-0912 and o1-preview-0912 clearly outperform all other models, followed by Claude-3.5-Sonnet and GPT-4o-0806. However, all LLMs evaluated are far from addressing our pattern recognition task; the best model, o1-mini-0912, only reaches an accuracy of less than $0.7$, while most models such as GPT-4o-0806 only achieve less than $0.4$ accuracy. Such performance indicates that the pattern recognition task still poses a significant challenge for most LLMs' in complicated tasks. Claude-3.5-Haiku achieves surprisingly low accuracy; upon checking examples, we find that the model often do not understand our prompt and see the target input as part of an incomplete data, thus refusing to answer the problem.

Interestingly, scaling up the number of shots is not always beneficial, similar to many tasks in Agarwal et al. [1]. For models other than Gemini, the performance drop over $512$ shots can be partly attributed to exceeding the 128K context limit [4]; however, for most language models evaluated (including o1 series), the performance growth often stops at no more than $256$ shots, where the context limit is not reached. Such issue stems from attention dispersion as stated in Yuan et al. [93]; as the number of examples increases, the attention weights which should be cast on the most informative shots is distracted by the less informative ones instead of lack of information retrieval ability. We validate this via ablation in Sec. 4.3.

## 4.2 MIR-Core: Problems Requiring Many-Shot

**Ablation on possible factors.** While we have obtained many pattern recognition problems, not all of them necessarily benefit from many-shot ICL; for example, a simple function such as adding two numbers or absolute value can be induced in a few shots. To study the inductive reasoning problems whose difficulties are *distinctive* between few-shot and many-shot, and curate a high-quality many-shot benchmark, we perform a detailed ablation study on possible factors for such distinctiveness. To better study such property, we define the following metric $D$:

$$D = \frac{D_1 + D_2}{2}, \text{ where } D_1 = \left[\frac{\text{acc@64} + \text{acc@128}}{2}\right] - \left[\frac{\text{acc@16} + \text{acc@32}}{2}\right],$$

$$D_2 = \left[\frac{\text{acc@32} + \text{acc@64} + \text{acc@128}}{3}\right] - \left[\frac{\text{acc@4} + \text{acc@8} + \text{acc@16}}{3}\right], \quad (1)$$

---

[4]Which only happens in $\leq 1\%$ case for 1024 shots but more common for 2048 shots. See Tab. 8 for details.

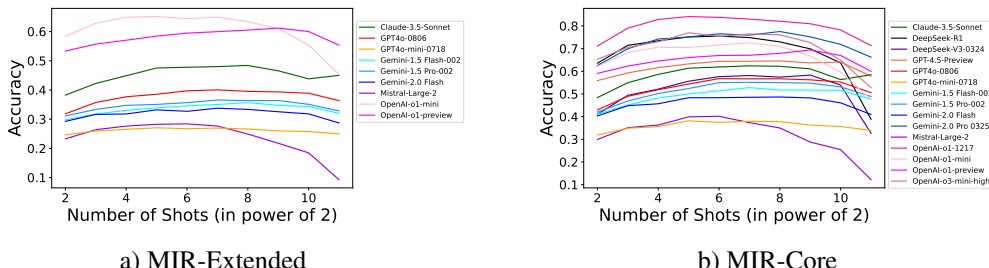

| a) MIR-Extended | b) MIR-Core |

Figure 2: The performance of LLMs on MIR-Extended (panel (a)) and MIR-Core (panel (b)). For better readability, we only show the most representative models; see Fig. 8 in Appendix D.2 for the rest. The benchmark poses challenge to all models tested. Most models will "saturate" at a particular number of shots, i.e., their performances stop to improve when more shots are given due to attention dispersion (See Sec. 4.3 for ablation).

where acc@$x$, $x \in \{4, 8, 16, 32, 64, 128\}$ is the average accuracy of {GPT-4o-0806, GPT-4o-mini-0718, Gemini-Pro 1.5-002, Gemini-Flash 1.5-002, Mistral-Large-2} at $x$-shot over 10 test cases.

Intuitively, $D$ is a combination of two components $D_1$ and $D_2$, each measures average performance growth from different few-shot to many-shot ranges; The range of $x$ is based on prior inductive reasoning work [11] and the number of shots where performances saturate on MIR-Extended. Ideally, we want to identify the factors which are positively related to $D$, and curate MIR-Core with problems having higher values of $D$.

With such metric $D$, we consider the following factors that are potentially relevant to the distinctiveness between few-shot and many-shot: **1) Ground truth function complexity:** 64-shot accuracy, function code length, LLM-evaluated function difficulty leveland problem topics; **2) Answer complexity:** number of different answers across 20000 shots, and the ratio of the most common answer out of 20000 shots; **3) Input complexity:** input length per shot.

As we aim to ensure the diversity of our evaluation, we did not select problems based on problem topics (See Appendix E.4 for ablation on problem topics). For the rest of the factors, we fit the ground-truth metric $D$ using a quadratic function with these factors (after normalization) as self-variables. We use quadratic function as we found some factors (e.g. # different answers), are roughly raised at both ends and concave in the middle, while others are roughly monotonic (e.g. code length); see Fig. 11 in Appendix E.5 for details.

The coefficients are illustrated as Fig. 3. as the result shows, ground truth function complexity is the dom-

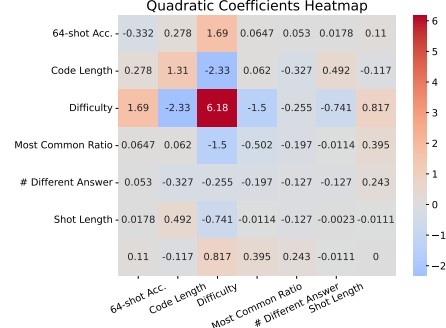

Figure 3: The coefficients of the quadratic function fitting $D$ with the aforementioned factors normalized between $[0, 1]$. The blank row and column are for constant factors. LLM-labeled difficulty is the leading factor for $D$, while answer diversity and shot length are less important.

inating factor for distinctiveness between few-shot and many-shot performance, among which LLM-labeled difficulty is a leading, positive factor (i.e. more difficult problem will require more shots). Answer diversity and input complexity are relatively less important. See Appendix E.5 for single-factor analysis, and Appendix E.3.2 for qualitative analysis on how difficulty affects $D$.

Another finding worth noting is that we find it highly non-trivial to get a reliable problem difficulty estimation from LLM: *LLM tend to underestimate inductive reasoning difficulty when given a simple underlying function.* For example, consider the one-line function that filters an interger list:

```
lambda l:  return [l[i] for i in range(1, len(l)) if l[i] % i == 0],
```

which gives [-4, 16, -63, -32, -5] as output when given input [48, -4, 16, -63, -32, -5, -32, -45]. GPT-4o-0806 gives difficulty score of 5 out of 10, indicating that this is a moderate question. However,

this question is so difficult that we have to exclude it from our benchmark, i.e. have 0 accuracy across all models for all numbers of shots mentioned in Sec. 3.2. To avoid LLM being tricked by the simple underlying function, we propose a multi-round conversation framework with self-reflection. In this framework, We first let the LLM to try to solve the problem without code by itself, and then reveal the ground-truth answer and let LLM to score the difficulty based on self-reflection. We found that with such framework, the evaluation from LLMs are much more accurate; see Appendix F.5 for details.

**Selection of data for MIR-Core.** We adopt the quadratic function's fitting result and select the $300$ problems with the highest predicted $D$-value as MIR-Core, each has $10$ test cases. To achieve a balance between achieving higher $D$-value for MIR-Core and unbiased evaluation for the LLMs involved in computing $D$-value, we do not use the problems with the highest ground truth $D$.

**Results on MIR-Core.** The evaluation results on MIR-Core are illustrated in Fig. 2(b). We again evaluate all 15 LLMs in Sec. 4.1 on MIR-Core. While the performance difference between few-shot and many-shot are more distinctive as expected, the relative performance and many-shot saturation phenomenon remain unchanged. We also evaluate the following six more cutting-edge LLMs: {OpenAI-o1-1217, OpenAI-o3-mini-high, DeepSeek-R1, DeepSeek-V3-0325, GPT-4.5-Preview, Gemini-2.0 Pro 0325}, many of which are models with long Chain-of-Thought (CoT) [86] process, i.e., "thinking" models.[5] Interestingly, while stronger models such as OpenAI-o1-1217 generally have higher performance, the saturation phenomenon persists.

### 4.3 Results with Duplicated Few-shots

To study whether the saturation of many-shot in Sec. 4.1 and 4.2 comes from the inability of retrieving the most useful shots for induction or the inability of aggregating many pieces of different, useful information, we conduct an ablation where we test {GPT-4o-0806, GPT-4o-mini-0718, Gemini-Pro 1.5-002, Gemini-Flash 1.5-002, Mistral-Large-2} on MIR-Core with 16-shot, but with the following two settings: **1) one shot duplicated** until total shots number of reach $\{16, 32, 64, 128, 256, 512, 1024, 2048\}$, while other 15 shots only appear once; and **2) all** 16 **shots duplicated** for $\{1, 2, 4, 8, 16, 32, 64, 128\}$ times.

The result is shown in Fig. 4, where solid lines are for original results on MIR-Core from Sec. 4.2, dashed lines are for scenario 1 (one shot duplicate), and dotted lines are for scenario 2 (all shots duplicate).When the number of shots increase, as shown in panel (b), the performance difference between normal many-shot and both scenario 1 and 2 increases, which indicates that LLMs can indeed aggregate many pieces of information from more shots and acquire performance gain (which is almost not the case for Mistral-Large-2, and thus its "saturation point" of performance with more shots is the lowest). However, the difference diminishes when there are more than $512$ shots (note this also applies for Gemini with 2M context length, thus this is not a problem of hard context limit). Such result indicates that too many pieces of information may actually harm LLMs' performance by distraction. Also, the performance of the dotted line (all shots duplicate) is in general not higher than that of the dashed line (one shot duplicate), which indicates that **the saturation problem is not in information retrieval, but from distraction when aggregating too many information**, as the two scenarios contain the same amount of information but the latter has higher difficulty for information retrieval.

### 4.4 Inductive Reasoning vs. Transductive Reasoning

In our previous results in Sec. 4, we did not instruct the model to include Chain-of-Thought (CoT) [86]; thus, the models can either conduct inductive learning with CoT or transductive learning by directly outputting solution for the target input. In this section, we study the performance difference between inductive and transductive performance of LLMs.

**Statistics in main results.** We first count the number of answers with and without CoT[6] in MIR-Core results (Sec. 4.2) and their respective correct rate; surprisingly, we find that in all 21 models, including long thinking models such as o1, answers without CoT (i.e. transductive results) have unanimously and significantly higher accuracy than those with CoT (i.e. inductive results). Tab. 2 shows the result for the most representative models; see Appendix D.4 for full results.

---

[5]Due to cost limit, we did not evaluate them in many of our other experiments.

[6]We count answers with $\geq 20$ characters before the final "Output:" as the ones with CoT.

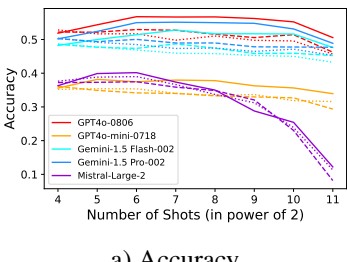
a) Accuracy

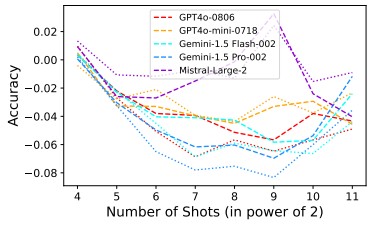
b) Accuracy difference

Figure 4: Results of duplicating shots. For panel (a), solid lines are for results on MIR-Core from Sec. 4.2, dashed lines are for scenario 1 (one shot duplicate), and dotted lines are for scenario 2 (all shots duplicate). Panel (b) is the result of dashed and dotted line subtracting solid line in panel (a).

Table 2: The results on MIR-Core of each model with CoT (inductive) and without CoT (transductive). See Tab. 9 for full results. Results are averaged over $\{4, 8, 16, 32, \ldots, 2048\}$-shot. The ratio of answer with and without CoT does not add up to $100\%$, as we did not count results where we are unable to extract answer. The result shows that while the preference of inductive vs. transductive varies, the performance of transductive reasoning is unanimously higher.

| Model | Answer w./ CoT (%) | Accuracy w./ CoT | Answer w./o. CoT (%) | Accuracy w./o. CoT |
|---|---|---|---|---|
| Claude-3.5-Sonnet | 98.73 | 0.585 | 1.08 | 0.775 |
| Gemini 1.5-Flash-002 | 20.11 | 0.306 | 79.75 | 0.539 |
| Gemini 1.5-Pro-002 | 20.96 | 0.339 | 78.90 | 0.561 |
| Gemini 2.0-Flash | 24.43 | 0.363 | 74.59 | 0.498 |
| GPT-4o-0806 | 10.85 | 0.488 | 88.34 | 0.540 |
| GPT-4o-mini-0718 | 37.21 | 0.279 | 61.77 | 0.414 |
| Mistral-Large-2 | 75.66 | 0.306 | 21.64 | 0.403 |
| o1-mini-0912 | 2.54 | 0.334 | 93.63 | 0.696 |
| o1-preview-0912 | 56.71 | 0.588 | 40.88 | 0.797 |
| DeepSeek-R1 | 9.69 | 0.298 | 85.85 | 0.757 |

**Are LLMs better transductive reasoners or better inductive reasoners?** To further validate whether the performance difference comes from inductive reasoning or from problems with different difficulty levels (e.g. LLMs only apply inductive reasoning on difficult problems), we further test MIR-Core with different prompts under two settings: 1) **forced CoT**, where the models are **forced** to write CoT; 2) **no CoT**, where the models are required to **not** write CoT and directly give answer. We evaluate all 15 models in Sec. 4.1 and DeepSeek-R1. See Appendix F.2 for prompts.

Fig. 5(a) shows the result of the most representative models on MIR-Core; see Appendix D.4 for other LLMs. For long CoT models such as o1 series and DeepSeek-R1, the performance of forced CoT is similar or slightly better than no CoT, indicating that such models prefer to present CoT for more difficult questions and hide CoT in the final answer for easier questions. However, for all other models (except GPT-4o-mini-0718), forced CoT indeed works worse than no CoT, and such performance gap increases with the number of shots (See Fig. 9 in Appendix D.4 for more results). That being said, the performance gap is smaller than those reported in Tab. 2, indicating that these models also prefer inductive reasoning for more difficult questions.

**Is CoT harmful for most LLMs in pattern recognition tasks?** The performance gap between no CoT and forced CoT for most LLMs seemingly leads to a counter-intuitive conclusion that CoT *harms* LLMs' performance in many-shot pattern recognition tasks. To verify whether this is the case, we conduct another ablation where compare the forced CoT results with another setting: **forced nonsense**, where the model is asked to first output a random paragraph of 700 characters [7], then conduct transductive reasoning and directly give its answer. We test {GPT-4o-0806, GPT-4o-mini-0718, Gemini-1.5 Flash-002, Gemini-1.5 Pro-002, Gemini-2.0 Flash, Mistral-Large-2} on MIR-Core for this experiment. The result is illustrated in Fig. 5(b). Unsurprisingly, CoT indeed *helps* LLMs' performance as forced CoT results are generally much better than forced nonsense results, and stronger models benefit more from CoT. However, such benefit weakens with more shots, indicating that CoT have yet to scale with many-shots to integrate thousands of pieces of information together.

---

[7]Average CoT length in forced CoT for models tested in this experiment.

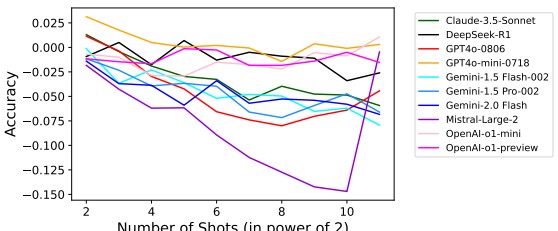 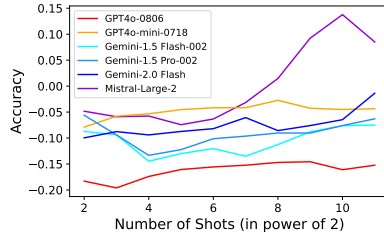

a) Acc. difference (forced CoT - no CoT)          b) Acc. difference (forced nonsense - CoT)

Figure 5: Panel (a) shows the accuracy difference of **forced CoT** and **no CoT** on MIR-Core, and panel (b) shows the difference between **forced nonsense** and **forced CoT**. The result shows that for most LLMs, the structural coherence between input-output pairs preserved during normal transductive reasoning prevails. CoT indeed helps reasoning, but its effect weakens with many shots and cannot compensate for breaking input-output format, especially with more shots.

**Conclusion.** To explain why CoT helps reasoning but LLMs still do better in transductive reasoning (no CoT) than inductive reasoning (forced CoT), we hypothesize such phenomenon comes from CoT breaking the **structural coherence** between input-output pairs. For example, consider a problem with two integers $a$ and $b$ as input and $\max(a, b)$ as output; transformers can easily duplicate the mapping relation between the three sets of tokens $a$, $b$ and $\max(a, b)$ as if going through a gradient descent with regression loss on examples as the training set, as suggested by many theoretical works in ICL [14, 82, 52]. However, a mapping from input to CoT makes the equivalent of gradient descent much more opaque, thus breaking the structural coherence that allows LLMs to "implicitly regress" through its attention matrix. The benefit of keeping structure coherence outweighs CoT, which explains why "forced nonsense" as transductive learning but without structural coherence works the worst, and why the performance gap between forced CoT and no CoT widens with more shots - the implicit regression effect gets stronger with more shots and consistent format.

## 5   Discussion and Conclusion

In this paper, we propose MIR-Bench, a novel, large-scale many-shot in-context pattern recognition reasoning benchmark and poses a difficult challenge for LLMs. We test $21$ LLMs from $4$-shot to $2048$-shot on our benchmark, and conduct extensive ablations on many aspects such as CoT, inductive vs. transductive, robustness, coding, RAG and meta-shot paradigm in addressing inductive reasoning problems. With many important insights concluded from our experiments, we believe our work provides a unique way of understanding LLM's intelligence level under long-context scenario.

**Downstream tasks of interest.** Beyond better understanding of LLM intelligence in general, our work is also more directly beneficial for several downstream tasks. Here we list two examples:

*Decision-making agents.* When making-decisions, online interaction can be costly and dangerous (e.g. controlling a robotic arm, or navigating Amazon and buy items). A good LLM agent should be able to learn from the past interactive experiences in the environment, either by building the back-ground dynamics model (e.g. model-based reinforcement learning [55]) or imitating past expert behavior (e.g. imitation learning [94]). Our benchmark fits into this type of application as the input can be seen as state-action pairs and the output can be seen as the outcome (utility gained and new states). See LMAct [67] for an example application.

*Programming-by-Example (PbE).* PbE is a long-studied program synthesis paradigm where the LLM needs to write code based on input-output examples, which is widely used in coding assistant [13] and Excel sheets autofill [21]. While our main evaluation does not involve writing code, we evaluate such application in Appendix C.2; with minor modification, our benchmark can serve as a solid basis for the downstream PbE works, such as Wei et al. [85].

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

# Appendix: MIR-Bench: Can Your LLM Recognize Complicated Patterns via Many-Shot In-Context Reasoning?

The appendix is organized as follows. First, we state limitations, future work and broader impact in Sec. A. Then, we discuss more related fields to our work, and conduct an extended comparison to all related many-shot ICL or inductive reasoning works to further illustrate the position of our work in Sec. B. After this, we provide additional empirical study results in Sec. C (which are also main results of this paper but postponed to the appendix due to page limit), then extra results and auxiliary ablations in Sec. D. Then, we provide statistical features of MIR-Bench in Sec. E. Finally, in Sec. F, we introduce more details in our experiments, including the prompts we adopted in our curation of dataset and ablation experiments and the regex rule we used for extracting the answer.

We hereby summarize the important novel insights obtained from experiments in the appendix:

1. LLMs are quite robust against errorneous shots in many-shot inductive reasoning tasks. (Sec. C.1)

2. The first-coding, then-running paradigm are not always scalable to many-shot case. Many-shot in-context pattern recognition remains an open problem. (Sec. C.2)

3. RAG is not a effective solution for addressing saturation issue of the many-shot pattern recognition task. (Sec. C.3)

4. It still remains an open challenge for LLMs to learn "meta-skills" of inductive reasoning from out-of-domain demonstrations. (Sec. C.4)

5. The evaluation on our benchmark is robust across different random seeds; i.e., the standard deviation of the performance is low. (Sec. D.1)

6. The performance of LLMs against erroneous shot largely depends on the ratio of errorneous shots; under the same ratio, the total number of shots does not change much. (Sec. D.5)

7. While generally adding more shots increases LLM's inductive performance, the performance change varies with problem types. LLMs improve the most on string manipulation tasks where each character in the input serves as a "shot" inside each example, and will not improve if the functions are too straightforward or too difficult. (Sec. E.4)

8. LLMs tend to underestimate inductive reasoning difficulty during evaluation given a concise ground truth. A better choice is to do a multi-round evaluation where LLMs can better evaluate difficulty by self-reflection on its attempt for solving the problem. (Sec. F.5)

## A    Limitations, Future Works and Broader Impact

**Limitations and future works.** First, to curate MIR-Core with problems that requires many-shot ICL, we studied many related factors such as types of problem and difficulty of the problems; however, they are not decisive enough. A more explainable rule for determining whether a problem needs many-shot would be an interesting avenue for future many-shot ICL works. Second, our test of pattern recognition is limited to text; it would be interesting for future work to explore the intelligence of multimodal models [78, 53, 12]. Third, while we have largely reduced the ambiguity of the underlying functions by filtering out those with insufficiently diverse input-output patterns and unsolvable by LLMs, some underlying functions could still be non-unique given our input-output pairs. Finally, our empirical studies have disproved some possible fixes to the saturation issue of many-shot ICL such as RAG, but do not provide a panecea. According to prior work [45], we hypothesize that supervised finetuning and/or reinforcement learning with in-context learning data would be a promising avenue to explore.

**Broader Impact.** Our work proposes an interesting and useful challange for LLM's long-context reasoning ability, and summarized many useful insights for future LLM studies. As we mentioned in the paper, our work is a step towards generalist AI agents that perceive the world from interaction examples and make decisions from demonstrations. Thus, our work inherently shares the societal impact with all other LLM papers: while LLMs could significantly boost human's working efficiency and production power of the society, the misuse of LLMs could cause harm to humans such as displacement of human workers.

# B Extended Related Work

**Programming-by-Examples (PbE).** PbE [58, 13] is a classic programming paradigm where programs are automatically written with user-provided input-output pairs as examples; it can be seen as an application of inductive reasoning in coding, and has wide application in sheet processing [21], data parsing [40], and systematic drawing [9]. It is traditionally addressed by symbolic-based approaches, such as heuristic search [21, 9], version space algebra [38] and learning weights for rule probabilities [54]; this symbolic formulation has largely limited the generalizability of PbE. Recently, as LLMs have proved themselves to be strong coders [22], several works tried to address general-purpose PbE with LLMs [71, 72, 70, 47]. None of them, however, considers many-shot scenario with more than 10 shots. Compared to existing works, Our benchmark is organized in a way that resembles many-shot PbE paradigm, but for most of the evaluations, the LLMs we tested are not required to write code; instead, they only need to directly predict output for new input. That being said, with minimal adaptation, our proposed benchmark can fill in the blank of many-shot PbE study (and we explored this in Sec. C.2).

**Extended comparison with literature.** Tab. 3 shows a detailed comparison of our work with existing works (including empirical study and benchmarks) in the field of many-shot and pattern recognition task. As shown in the table, our work is indeed unique among all the many-shot ICL and inductive reasoning works.

# C More Empirical Studies on MIR-Bench

## C.1 Robustness of LLM Inductive Intelligence

While many works [1] have studied LLM's many-shot ICL performance, the robustness of LLM's many-shot ICL ability [98], i.e. the accuracy given incorrect examples, is still largely underexplored. In this section, we explore the performance change with increasing number of shots with incorrect answers.

**Evaluation Setup.** We test all 15 models in Sec. 4.1 on MIR-Core with 3 different settings: 1) the "unaware" setting, where the models do not know there are incorrect answers in the provided examples; 2) the "aware-error" setting, where the models know that some (unknown number of) examples are incorrect; and 3) the "aware-ratio" setting, where the models know exactly how many shots are incorrect out of all given shots. The three settings are mostly the same, with slight difference in prompt; see Appendix F.2 for details. We test $\{64, 256, 1024\}$ shots $\times$ error ratio of $\{1/64, 1/32, 1/16, 1/8, 1/4, 1/2, 3/4\}$ respectively. See Appendix F.7 for data generation details.

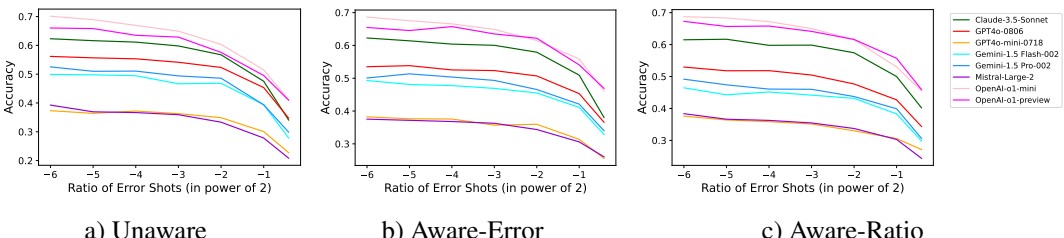

|  a) Unaware | b) Aware-Error | c) Aware-Ratio |

Figure 6: The accuracy of representative models with erroneous shots under different prompt settings with 64 shots (see Fig. 10 in Appendix D.5 for full results). The result shows LLMs are generally quite robust against erroneous shots.

**Results.** The results for 64-shot are illustrated in Fig. 6 (see Appendix D.5 for the rest). Surprisingly, we found that LLMs are generally quite robust against errorneous shots; their performance are only slightly harmed below $1/8$, and can maintain decent performance even with $3/4$ error rate. We find that generally, there is generally no significant performance difference in different awareness level of erroneous shots; some exceptions are Gemini-2.0 Flash and Claude-3.5-Haiku (see Fig. 10 in Appendix D.5), where the answering paradigm of the former remains the same, and the latter accepts the target input as part of the "incomplete" data and rejects answering questions less frequently. **Overall, LLMs are quite robust against errorneous shots in many-shot inductive reasoning tasks.**

Table 3: The topic, validity and reproducibility comparison between our benchmark and the most related prior many-shot / long-context benchmarks in the first part, and pattern recognition (inductive and transductive) reasoning benchmarks in the second part. To save space, we abbreviate "Many Shot" as MS, "Pattern Recognition" as PR (△ represents "classification only"), "Prob." as problems, and "I/O Div." as "Input/Output Diversity" (**having at least** 2 **different input-output types, e.g., given an array and output an integer, or given a pair of strings and output a choice**). "Gen." means "Generative", which means whether new test cases can be easily generated without much human effort. "LB" means whether a leaderboard is available, and "EE" means "Easy Evaluation", i.e., whether a pipeline for evaluating any given new model exists. "New Data" means whether the input-output data never appears in existing benchmarks; if so, the benchmark is not a compilation of existing dataset and is secured against data contamination. Note, the counting of #PR Problems and "Gen." take different target input-output for the same function into account, but **do not take different sets of shots into account**.

| Evaluations | MS | PR | # PR Prob. | I/O Div. | Max # Shots | Gen. | LB | EE | New Data |
|---|---|---|---|---|---|---|---|---|---|
| Classifications [45] | ✓ | △ | ~25K | ✗ | 2000 | ✗ | ✗ | ✗ | ✗ |
| Many-Shot ICL [1] | ✓ | ✓ | 450 | ✓ | 2048 | ✓ | ✗ | ✗ | △ |
| Classifications [6] | ✓ | △ | 1250 | ✗ | 2000 | ✗ | ✗ | △ | ✗ |
| Visual Classifications [29] | ✓ | △ | 4010 | ✗ | ~2000 | ✗ | ✗ | ✓ | ✗ |
| Instruction Following [98] | ✓ | ✗ | 0 | ✓ | 300 | ✓ | ✗ | ✓ | ✗ |
| 2D Classifications [99] | ✓ | △ | 100 | ✗ | 256 | ✓ | ✗ | △ | ✓ |
| LLM Judge [75] | ✓ | ✗ | 0 | ✗ | 512 | ✓ | ✗ | ✗ | ✓ |
| HELMET [92] | ✓ | △ | 500 | ✓ | ~10K | ✗ | ✗ | ✓ | ✗ |
| LongICLBench [46] | ✓ | △ | 3000 | ✗ | ~2000 | ✗ | ✓ | ✓ | ✗ |
| ManyICLBench [101] | ✓ | △ | 1000 | ✓ | 7252 | ✗ | ✗ | ✓ | ✗ |
| LMAct [67] | ✓ | ✗ | N/A* | ✗ | 256 | ✓ | ✗ | ✓ | ✓ |
| LongBench [4] | ✓ | △ | 400 | ✓ | 600 | ✗ | ✓ | ✓ | ✓ |
| BABILong [36] | ✗ | ✓ | unknown | ✓ | unknown | ✓ | ✓ | ✓ | ✓ |
| KORBench [51] | ✗ | ✓ | 50 | ✓ | 3 | ✗ | ✓ | ✓ | ✓ |
| SolverLearner [10] | ✗ | ✓ | 1300 | ✓ | 16 | ✓ | ✗ | ✗ | ✓ |
| Case2Code [70] | ✗ | ✓ | 1.3M | ✓ | 10 | ✗ | ✗ | ✗ | ✓ |
| DEER [91] | ✗ | ✓ | 1250 | ✗ | 3 | ✗ | ✗ | ✗ | ✓ |
| List functions [66] | ✗ | ✓ | 4000 | ✗ | 5 | ✓ | ✗ | ✓ | ✓ |
| SyGus [84] | ✗ | ✓ | 89 | ✓ | 3 | ✗ | ✗ | ✓ | ✓ |
| ARC [11] | ✗ | ✓ | 800 | ✗ | 3 | ✗ | ✓ | ✓ | ✓ |
| 1D-ARC [88] | ✗ | ✓ | 900 | ✗ | 3 | ✗ | ✗ | ✓ | ✓ |
| Mini-ARC [32] | ✗ | ✓ | 150 | ✗ | 3 | ✗ | ✗ | ✓ | ✓ |
| WILT [5] | ✗ | ✓ | 50 | ✗ | 30 | ✗ | ✓ | ✓ | ✓ |
| LogicVista [87] | ✗ | ✓ | 107 | ✓ | 10 | ✗ | ✗ | ✓ | ✓ |
| CLUTRR [73] | ✗ | ✓ | 70K | ✗ | N/A | ✓ | ✗ | ✓ | ✓ |
| MIRAGE [43] | ✗ | ✓ | 2000 | ✓ | 8 | ✓ | ✗ | ✗ | ✓ |
| ACRE [96] | ✗ | ✓ | 30K | ✗ | 10 | ✓ | ✗ | ✗ | ✓ |
| Mini-SCAN [65] | ✓ | ✓ | 400 | ✗ | 100 | ✓ | ✗ | ✓ | ✓ |
| **Ours** | ✓ | ✓ | 6930 | ✓ | 2048 | ✓ | ✓ | ✓ | ✓ |

* LMAct has only a few tasks, but it is interactive and thus hard to count the number of problems.

## C.2 SolverLearner: Is "First-Coding, Then-Running" the Cure?

For better inductive reasoning ability, Cheng et al. [10] proposed SolverLearner, an inductive reasoning framework where LLMs write code first for inductive reasoning problems and then generate answers with python interpreter. With such framework, the authors claim that LLMs demonstrate remarkable inductive reasoning capabilities under their framework. However, their study is limited to a few relatively weak LLMs, (GPT-3.5, GPT-3), limited amount of inductive reasoning problems and few-shot; to check whether such solution also works for the many-shot pattern recognition task, we re-implement their method on MIR-Core (see Appendix F.8 for prompts).

We test SolverLearner with {DeepSeek-R1, Claude-3.5 Sonnet, GPT-4o-0806, GPT-4o-mini-0718, Gemini-Flash 2.0, Gemini-Pro 1.5-002, Gemini-Flash 1.5-002, Mistral-Large-2} for {16, 64, 256, 1024} shots respectively on MIR-Core. For each code snippet generated by LLMs, we set a limit of 1 second for execution, as we need to run 300 functions $\times$ 10 test cases $\times$ 4 different number of shots $\times$ 8 models = 96000 code snippets.

Table 4: The accuracy at 16, 64, 256 and 1024 shots respectively for SolverLearner on MIR-Core, and its performance difference from results reported in Sec. 4.2. We plot improvements with $> 0.02$ **blue** and $< -0.02$ **red**. We find that the performance of SolverLearner varies from model to model, and does not necessarily perform better than normal inductive reasoning paradigm. Also, SolverLearner curves under many-shot are more "flat"; i.e., it does not seem to make good use of extra information from a large number of shots. Such result indicates that LLM many-shot inductive reasoning is still an open problem, and straightforward solutions such as SolverLearner are not suffice yet.

| Model | Acc.@16 | Acc.@64 | Acc.@256 | Acc.@1024 |
|---|---|---|---|---|
| DeepSeek-R1 | 0.756(+0.022) | 0.762(+0.007) | 0.748(+0.018) | 0.640(+0.003) |
| Claude-3.5 Sonnet | 0.577(-0.009) | 0.604(-0.015) | 0.605(-0.017) | 0.603(+0.04) |
| GPT4o-0806 | 0.530(+0.012) | 0.534(-0.033) | 0.538(-0.029) | 0.556(+0.004) |
| GPT4o-mini-0718 | 0.350(-0.006) | 0.375(+0.003) | 0.386(+0.008) | 0.370(+0.014) |
| Gemini-2.0 Flash | 0.469(+0.066) | 0.487(+0.003) | 0.493(+0.006) | 0.487(+0.026) |
| Gemini-1.5 Pro-002 | 0.469(-0.029) | 0.495(-0.055) | 0.483(-0.067) | 0.491(-0.04) |
| Gemini-1.5 Flash-002 | 0.473(-0.009) | 0.484(-0.03) | 0.479(-0.038) | 0.486(-0.03) |
| Mistral-Large-2 | 0.420(+0.057) | 0.430(+0.028) | 0.428(+0.078) | 0.356(+0.102) |

Tab. 4 demonstrates the accuracy of each model (with difference from the standard results reported in Sec. 4.2) on MIR-Core, and Tab. 5 demonstrates the error rate when writing code. We found that the effect of SolverLearner varies from model to model; i.e., SolverLearner does not necessarily improve performance on our benchmark. Also, SolverLearner does not seem to utilize many-shot well; the performance increase from 16-shot to 1024-shot is much smaller than that of standard performance reported in Sec. 4.2. We hypothesize such issue, similar to that in Sec. 4.4, stems from the complicated nature of the code. Moreover, models with relatively weaker long-context ability, such as Mistral-Large-2, has much higher error rate with many-shot as the context length goes beyond its "effective" [26] context length; DeepSeek-R1 as a long CoT model also struggles with high runtime error rate from many-shot inductive reasoning. Thus, many-shot pattern recognition is still an open problem and not yet solved by straightforward solutions such as SolverLearner. The insight can be summarized as follows:

## C.3 Can RAG Help Many-Shot Pattern Recognition?

One possible way to bypass the problem of many-shot saturation is Retrieval Augmented Generation (RAG) [19]; i.e., instead of feeding every given shot into the LLM and disperses the model's attention, we only select a few shots that are the most related to the target input, thus forcing the model to concentrate on the few but useful shots in its context. Usually, there are two prevalent ways to select such shots: selected by LLM [39] or selected by embedding [41, 89]. The former is infeasible in our many-shot pattern recognition task, as each of our shot is already very precise and hard be further compressed by LLM as in other RAG works [39]; also, selection of shots with over 2000 candidates for each of the 3000 test cases in MIR-Core will be prohibitively expensive and/or error-prone for LLMs. Thus, we will focus on embedding-based RAG for this part.

Table 5: The Do-Not-Finish (i.e., no solution function generated) and Runtime Error (RE, including timeout and exception during running) rate at 16, 64, 256 and 1024 shots respectively for Solver-Learner on MIR-Core. Generally, with more shots, the error rate of LLMs will increase. Some models such as DeepSeek-R1 and Mistral-Large-2 has high error rate under long context scenario.

| Model | DNF@16 | RE@16 | DNF@64 | RE@64 | DNF@256 | RE@256 | DNF@1024 | RE@1024 |
|---|---|---|---|---|---|---|---|---|
| DeepSeek-R1 | 0 | 0.0303 | 0.0003 | 0.0689 | 0.0003 | 0.1010 | 0.057 | 0.2061 |
| Claude-3.5-Sonnet | 0 | 0.0027 | 0 | 0.0063 | 0 | 0.0007 | 0 | 0.0037 |
| GPT4o-0806 | 0 | 0.009 | 0 | 0.0103 | 0 | 0.0157 | 0.0033 | 0.0137 |
| GPT4o-mini-0718 | 0 | 0.0103 | 0 | 0.0147 | 0 | 0.0167 | 0.0033 | 0.017 |
| Gemini-2.0 Flash | 0 | 0.0023 | 0 | 0.007 | 0 | 0.007 | 0 | 0.0068 |
| Gemini-1.5 Flash-002 | 0 | 0.0093 | 0 | 0.0117 | 0 | 0.0087 | 0 | 0.011 |
| Gemini-1.5 Pro-002 | 0 | 0.0093 | 0 | 0.008 | 0 | 0.009 | 0 | 0.0107 |
| Mistral-Large-2 | 0 | 0.008 | 0 | 0.0077 | 0.0047 | 0.012 | 0.1163 | 0.0473 |

**Evaluation setup.** We evaluate GPT-4o-0806, GPT-4o-mini-0718, Gemini-1.5 Pro, Gemini-1.5 Flash, Mistral-Large-2 on MIR-Core. To generate embedding vectors effectively, we choose a small but recognized sentence encoder, all-MiniLM-L6-v2 [81], to generate vectors for each shot. We test 128 to 2048 shots by selecting 64 shots with the closest (cosine similarity) vector representation to the target input, and compare it with 64 shots that are randomly sampled from the same 128 to 2048 shots.

**Results.** The result is illustrated in Tab. 6. The result shows no significant performance difference between RAG and randomly selecting shots, thus disproving the effectiveness of embedding-based RAG.

Table 6: The performance comparison between selecting 64 shots using RAG and random selection for many-shot pattern recognition. There is no significant performance difference between the two strategies for selecting shots.

| # Shots | Selection | GPT-4o | GPT-4o-mini | Gemini-1.5-Pro | Gemini-1.5 Flash | Mistral |
|---|---|---|---|---|---|---|
| 128 | RAG | 51.77 | 38.50 | 51.53 | 46.33 | 27.20 |
| | random | 50.40 | 37.10 | 51.20 | 46.43 | 26.03 |
| 256 | RAG | 49.83 | 37.33 | 50.40 | 44.73 | 27.00 |
| | random | 50.17 | 37.27 | 50.83 | 45.63 | 27.37 |
| 512 | RAG | 49.33 | 37.17 | 50.33 | 45.20 | 27.53 |
| | random | 50.90 | 36.93 | 51.27 | 45.23 | 27.10 |
| 1024 | RAG | 49.60 | 37.60 | 50.77 | 45.67 | 26.67 |
| | random | 51.03 | 37.63 | 50.77 | 45.67 | 27.23 |
| 2048 | RAG | 50.53 | 39.27 | 51.75 | 46.67 | 26.67 |
| | random | 51.63 | 37.23 | 51.37 | 45.83 | 27.03 |

## C.4 Can LLMs Learn Inductive Skills from Out-of-Domain Meta-Shots?

Till now, we have mostly limited our many-shot experiments within *in-distribution* learning, which means all the given shots indicates the same function as the target input. A more desirable ability, however, is to learn from *out-of-domain* inductive reasoning traces: by given successful demonstrations on extracting rules from other examples, we hope LLMs to learn the "meta-skills" for pattern recognition (inductive reasoning in this case), e.g., to pick up a few examples, propose an assumption, and then verify with other examples (as explored by Wang et al. [84] with training).

**Evaluation Setup.** We test Gemini-1.5 Pro-002 and Gemini-1.5 Flash-002 on MIR-Core. For each problem, we select correct (test case, LLM answer)-pairs from GPT-4o-0806's output in Sec. 4.4 with 8-shot forced CoT, and filter out problems with invalid CoT by GPT-4o-0806. For each test case, we sample 4, 8, 16, 32 different (test case, LLM answer)-pairs, and put them before the original MIR-Core problem as meta-shots; each meta-shot is separated by a line of '==='. See Appendix F.9 for details on prompts. We test the result of {4, 16, 64, 256, 1024} in-distribution shots.

**Results.** The result is illustrated in Tab. 7. The result gives two insights: 1) the effect of meta-shots varies across models. For models like Gemini-1.5 Pro, meta-shots will slightly benefit CoT performance, but cannot fully bridge the gap between forced CoT (inductive) and no CoT (transductive); 2)

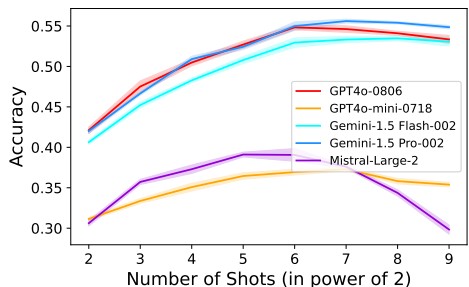

Figure 7: The performance of 5 cutting-edge LLM models on MIR-Extended with temperature $0.7$ across 5 runs. The result clearly shows that the standard deviation of accuracy is always below $0.01$, and thus the evaluation is highly stable.

the effect of meta-shot CoT slightly increases when given meta-shot CoT examples, but decreases when given more shots in the current problem, which is consistent with our finding in Sec. 4.4 that CoT struggles with more shots. **In general, LLMs have still yet to learn the "meta-skills" from out-of-domain demonstrations, which poses an interesting research topic for future.**

Table 7: The results of Gemini models with out-of-distribution meta-shots for inductive reasoning. Overall, more meta-shots leads to slightly better performance, but such effect weakens with more in-distribution shots and is not necessarily better than no meta-shots. Such result indicates that the models are yet to summarize and apply useful reasoning skills from in-context demonstrations.

|  | In-distribution #shots | 0(-meta-shots) | 4 | 8 | 16 | 32 |
|---|---|---|---|---|---|---|
| Gemini-1.5 Pro-002 | 4 | 38.67 | 39.17 | 41.33 | 41.97 | 42.97 |
|  | 16 | 45.73 | 47.63 | 47.90 | 48.63 | 48.53 |
|  | 64 | 49.97 | 48.10 | 49.67 | 50.57 | 49.80 |
|  | 256 | 49.43 | 49.17 | 49.53 | 49.43 | 50.27 |
|  | 1024 | 49.60 | 47.77 | 48.50 | 49.60 | 49.60 |
| Gemini-1.5 Flash-002 | 4 | 37.93 | 34.77 | 35.33 | 37.60 | 40.74 |
|  | 16 | 45.50 | 42.20 | 42.57 | 41.47 | 42.87 |
|  | 64 | 47.80 | 42.90 | 42.83 | 44.17 | 45.17 |
|  | 256 | 48.70 | 42.63 | 44.33 | 44.23 | 45.80 |
|  | 1024 | 46.10 | 40.23 | 41.40 | 42.07 | 43.90 |

# D  Complete Results and Auxiliary Ablations

## D.1  Analysis on the Stability of Evaluation

As we use $0$ temperature in the evaluations in our main papers, it is possible that the performance vary across LLM inferences with different random seeds, hence making our evaluation unreliable. To address such concern, we report the mean and standard deviation of the performance across 5 models {GPT-4o-0806, GPT-4o-mini-0718, Gemini-Pro 1.5-002, Gemini-Flash 1.5-002, Mistral-Large-2} over 5 different inferences with temperature $0.7$ in Fig. 7 on MIR-Extended with $4, 8, 16, 32, 64, 128, 256, 512$ shots. The result clearly shows that the standard deviation for all models are very small, and thus our evaluation is reliable.

## D.2  Complete Results on MIR-Extended and MIR-Core (Sec. 4.1, 4.2)

For better readability, we only put the performance of part of the models for MIR-Extended and MIR-Core in the main paper; Fig. 8 demonstrates the performance of all models.

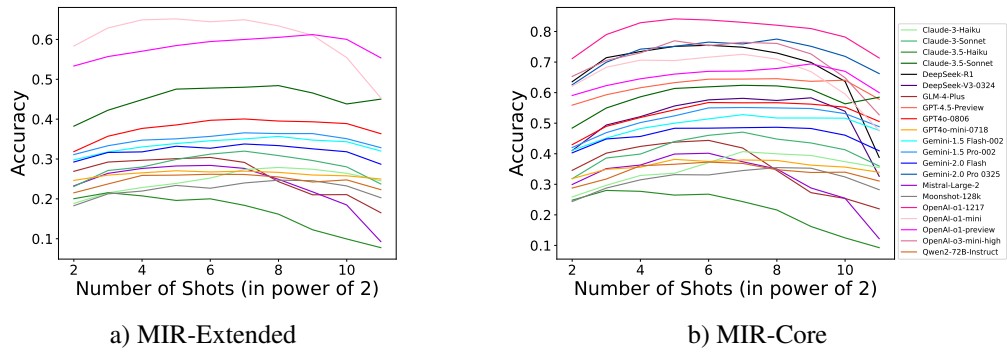

a) MIR-Extended                                    b) MIR-Core

Figure 8: The performance of all LLMs on MIR-Extended (panel (a)) and MIR-Core (panel (b)). As shown in Fig. 2 in the main paper, the benchmark poses challenge to almost all models tested. All models, including OpenAI-o1-1217, "saturate" at a particular number of shots, i.e., their performances stop to improve when more shots are given due to limited information integration capability.

## D.3  Out-of-Context Rate for 1024 and 2048 shot in MIR-Extended and MIR-Core

Tab. 8 shows the rate of out-of-context error we received when invoking APIs for MIR-Extended and MIR-Core. Some models other than Gemini (which has $> 1M$ context length) have an error rate of $0$, which could due to its internal truncation.

Table 8: Out-of-context rate for model API calls on MIR-Core and MIR-Extended.

| Model | MIR-Core 1024-shot (%) | MIR-Core 2048-shot (%) | MIR-Extended 1024-shot(%) | MIR-Extended 2048-shot (%) |
|---|---|---|---|---|
| Claude-3-Haiku | 0 | 1.67 | 0 | 0.98 |
| Claude-3-Sonnet | 0 | 1.67 | 0 | 0.87 |
| Claude-3.5-Haiku | 0.96 | 2.4 | 0.65 | 1.37 |
| Claude-3.5-Sonnet | 0 | 1.77 | 0 | 0.97 |
| Gemini-1.5 Flash-002 | 0 | 0 | 0 | 0 |
| Gemini-1.5 Pro-002 | 0 | 0 | 0 | 0 |
| Gemini-2.0 Flash | 0 | 0 | 0 | 0 |
| GLM-4-Plus | 0 | 0 | 0 | 0 |
| GPT-4o-0806 | 0.33 | 5.67 | 0.14 | 4.47 |
| GPT-4o-mini-0718 | 0.33 | 5.67 | 0.17 | 4.47 |
| Mistral-Large-2 | 0.67 | 10.67 | 0.29 | 8.66 |
| Moonshot-128K | 0 | 0 | 0.19 | 0.14 |
| OpenAI-o1-mini-0912 | 1 | 11 | 0.58 | 8.80 |
| OpenAI-o1-preview-0912 | 1 | 11 | 0.58 | 8.80 |
| Qwen2-72B-Instruct | 0 | 0 | 0.37 | 8.29 |
| DeepSeek-R1 | 0 | 0 | N/A | N/A |
| DeepSeek-v3-0324 | 0 | 0 | N/A | N/A |
| Gemini-2.0 Pro-0325 | 0 | 0 | N/A | N/A |
| GPT4.5-Preview | 0.33 | 6.67 | N/A | N/A |
| OpenAI-o1-1217 | 0 | 2.67 | N/A | N/A |
| OpenAI-o3-mini-high | 0.37 | 6.67 | N/A | N/A |

## D.4  More Results on Many-Shot Inductive Reasoning vs. Deductive Reasoning

Tab. 9 lists the ratio of forced CoT (inductive reasoning) / no CoT (transductive reasoning) and their respective performance for more models on MIR-Core using original prompt. While the preference for inductive or transductive reasoning varies wildly across different models, the accuracy of transdutive reasoning is unanimously and significantly higher.

Fig. 9 illustrates the performance difference for more models between forced CoT and no CoT. The result shows that transductive reasoning results (with no CoT) are indeed better than inductive reasoning results (with forced CoT), and such gap increases with the number of shots.

Table 9: The results on MIR-Core of each model with and without CoT. Results are averaged over $\{4, 8, 16, 32, 64, 128, 256, 512, 1024, 2048\}$-shot. Error rate include cases where answer cannot be extracted and API error for exceeding context length. Claude-3.5-Haiku often refuses to answer the question due to "incomplete data".

| Model | Answer w./ CoT (%) | Accuracy w./ CoT | Answer w./o. CoT (%) | Accuracy w./o. CoT | Error (%) |
|---|---|---|---|---|---|
| Claude-3-Haiku | 51.03 | 0.278 | 47.66 | 0.441 | 1.31 |
| Claude-3-Sonnet | 20.46 | 0.233 | 76.98 | 0.475 | 2.56 |
| Claude-3.5-Haiku | 65.11 | 0.317 | 1.32 | 0.823 | 33.57 |
| Claude-3.5-Sonnet | 98.73 | 0.585 | 1.08 | 0.775 | 0.19 |
| Gemini 1.5-Flash-002 | 20.11 | 0.306 | 79.75 | 0.539 | 0.14 |
| Gemini 1.5-Pro-002 | 20.96 | 0.339 | 78.90 | 0.561 | 0.14 |
| Gemini 2.0-Flash | 24.43 | 0.363 | 74.59 | 0.498 | 0.98 |
| GLM-4-Plus | 19.70 | 0.248 | 79.33 | 0.388 | 0.97 |
| GPT-4o-0806 | 10.85 | 0.488 | 88.34 | 0.540 | 0.81 |
| GPT-4o-mini-0718 | 37.21 | 0.279 | 61.77 | 0.414 | 1.02 |
| Mistral-Large-2 | 75.66 | 0.306 | 21.64 | 0.403 | 2.70 |
| Moonshot-128K | 43.40 | 0.242 | 53.11 | 0.398 | 3.50 |
| o1-mini-0912 | 2.54 | 0.334 | 93.63 | 0.696 | 2.41 |
| o1-preview-0912 | 56.71 | 0.588 | 40.88 | 0.797 | 3.82 |
| Qwen2-72B-Instruct | 1.85 | 0.130 | 97.05 | 0.349 | 1.10 |
| DeepSeek-R1 | 9.69 | 0.298 | 85.85 | 0.757 | 4.46 |
| DeepSeek-v3-0324 | 10.40 | 0.329 | 84.54 | 0.570 | 5.06 |
| Gemini-2.0 Pro-0325 | 78.95 | 0.691 | 20.63 | 0.872 | 0.41 |
| GPT-Preview-4.5 | 35.94 | 0.543 | 63.29 | 0.669 | 0.77 |
| OpenAI-o1-1217 | 3.55 | 0.469 | 96.12 | 0.811 | 0.33 |
| OpenAI-o3-mini-high | 85.48 | 0.697 | 13.42 | 0.806 | 1.10 |

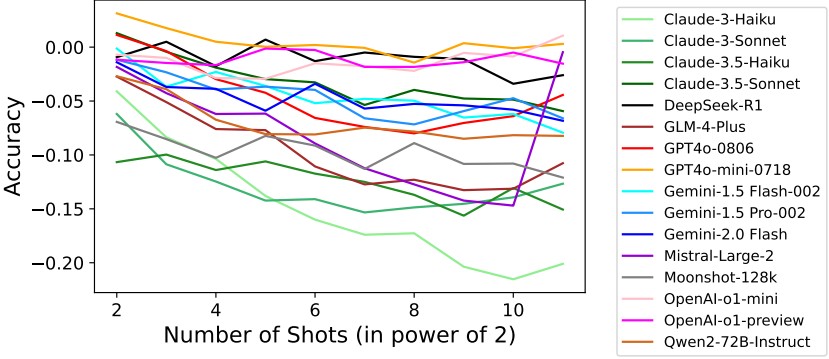

Figure 9: Performance difference for 16 LLMs on MIR-Core between forced CoT and no CoT. For long-CoT models (o1 series and DeepSeek-R1), forced CoT works similar or slightly better than no CoT, but the gain diminishes with more shots. For the rest of the models, forced CoT almost always works worse (with the exception of GPT4o-mini-0718), and such gap increases with the number of shots. Mistral-Large-2's gap decreases dramatically at 2048-shot as such setting often exceeds its context length and the performance is low under both settings.

## D.5 Complete Results on Robustness of LLM Inductive Intelligence

Fig. 10 shows the results of models on 64-shot, 256-shot and 1024-shot with different error rate for the shots, where the solid lines are 256-shot or 1024-shot accuracy respectively. We find that there are no significant performance difference across the same error rate with different number of shots (with the exception of o1-mini-0912 with 1024 shots), and the robustness persists across different number of shots.

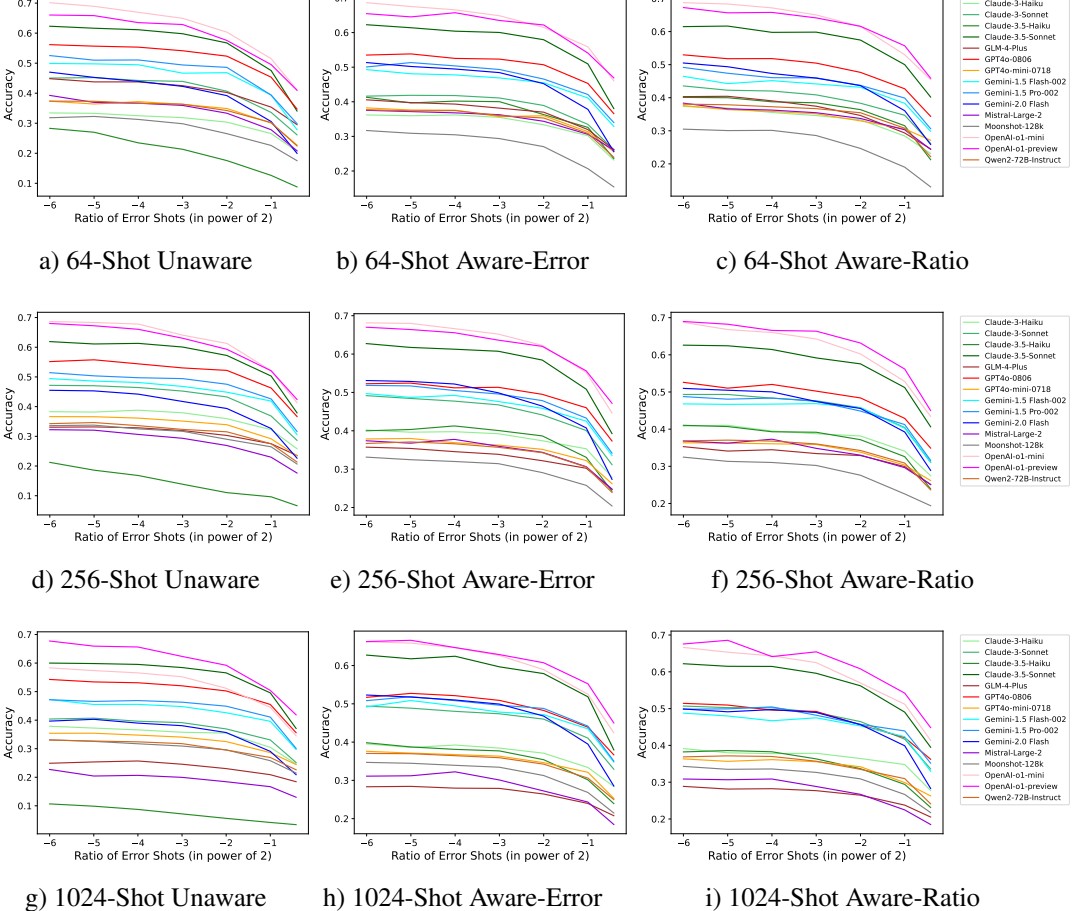

Figure 10: The accuracy of models with erroneous shots under different prompt settings. The performance of the same error rate with different numbers of total shots are similar.

# E    Statistical Features of MIR-Bench

## E.1    Data Source

Tab. 10 shows that out of 693 functions in MIR-Extended and 300 functions in MIR-Core, how many problems are extracted from each coding benchmark (HumanEval+, MBPP+ and APPS). The former two have Apache-2.0 licenses, and the latter has a MIT license.

Table 10: Number of functions extracted from each coding benchmark.

|              | HumanEval+ | MBPP+ | APPS | Total |
|--------------|------------|-------|------|-------|
| MIR-Core     | 26         | 35    | 239  | 300   |
| MIR-Extended | 53         | 89    | 551  | 693   |

## E.2    Input-Output Form

One advantage of MIR-Bench over existing works is that our curated problems have much more diverse input-output forms. To quantitatively illustrate this, we use GPT-4o-0806 to label the input-output forms. The result is illustrated in Tab. 12, which clearly shows the diversity of problems in our benchmark.

Table 11: Number of input-output forms (input -> output) in MIR-Extended and MIR-Core, sorted by instances. All data types (e.g. str, int) are in python format.

| Input-Output Form (MIR-Extended) | Count | Input-Output Form (MIR-Core) | Count |
|---|---|---|---|
| str -> str | 158 | str -> str | 86 |
| str -> int | 40 | int -> int | 18 |
| int -> int | 37 | list[float] -> list[float] | 11 |
| list[float] -> float | 24 | int, int -> int | 9 |
| int, int -> int | 24 | int -> str | 8 |
| str, str -> str | 22 | str, str -> str | 8 |
| list[int] -> int | 19 | str -> int | 8 |
| list[float] -> list[float] | 18 | list[int] -> int | 5 |
| list[float] -> int | 17 | str -> list[str] | 5 |
| int -> str | 13 | list[str] -> list[str] | 4 |
| str, int -> str | 10 | list[int] -> list[int] | 4 |
| float, float -> float | 10 | float -> int | 3 |
| list[int] -> list[int] | 10 | float -> float | 3 |
| str -> list[str] | 8 | list[float], list[float] -> list[float] | 3 |
| int, int -> list[int] | 6 | int, int, int -> int | 3 |
| float, float, float -> float | 6 | int, int, int -> str | 3 |
| int, int -> str | 6 | str, str -> int | 3 |
| str, str -> int | 5 | str -> list[int] | 3 |
| int, int, int -> int | 5 | list[float] -> float | 2 |
| float -> int | 5 | int, int -> str | 2 |
| list[float], float -> list[float] | 5 | list[str] -> int | 2 |
| float -> str | 5 | float -> str | 2 |
| others | 240 | others | 103 |

Table 12: Number of problems for each difficulty level labeled by LLM, normalized from 0 to 1.

| Difficulty Level | # MIR-Extended | # MIR-Core |
|---|---|---|
| 0.05 | 1 | 0 |
| 0.1 | 81 | 21 |
| 0.2 | 103 | 53 |
| 0.3 | 118 | 50 |
| 0.35 | 1 | 1 |
| 0.4 | 51 | 27 |
| 0.45 | 2 | 0 |
| 0.5 | 75 | 33 |
| 0.55 | 10 | 5 |
| 0.6 | 7 | 2 |
| 0.65 | 8 | 4 |
| 0.7 | 163 | 77 |
| 0.75 | 25 | 8 |
| 0.8 | 32 | 13 |
| 0.85 | 16 | 6 |

## E.3 Difficulty Level

### E.3.1 Problem Counts for Difficulty Levels

Tab. 11 shows the number of problems for each LLM-labeled difficulty level (see Sec. F.5 for details), which shows that the problems in MIR-Bench has diverse difficulty levels.

### E.3.2 Qualitative Analysis on the Effect of Difficulty Levels

In general, we find that the difficulty is positively correlated with the benefit of many-shot ICL (see Sec. E.5 for detailed numbers). Here, we append some examples of difficult and easy questions in our benchmark and analysis on whether they could be benefited from many-shot ICL:

```
def add(x: int, y: int): # Normalized difficulty level 0.1
    """
    Add two numbers x and y
    » add(2, 3)
    5
    » add(5, 7)
    12
    """
    return x + y
```

Apparently, such a function is extremely easy to induce, and thus few-shot ICL is sufficient; LLMs will not benefit much from many-shot ICL.

Here, we show a slightly harder question:

```
def find_sum(arr): # normalized difficulty level 0.4
    return sum(set(arr))
```

This question is slightly harder as it involves two operations: first remove all duplicate elements in a list, then get the sum. The model will need to look at multiple examples with and without duplicated elements to rule out other possible functions, e.g. sum of the array, fraction of the sum of array, etc. Thus, many-shot ICL will help more than the last function.

Finally, we give an example of a more difficult question:

```
def vowels_count(s): # normalized difficulty level 0.7
    """
    Write a function vowels_count which takes a string representing a word as input and returns the
    number of vowels in the string. Vowels in this case are 'a', 'e', 'i', 'o', 'u'. Here, 'y' is also a vowel, but
    only when it is at the end of the given word.
    Example:
    » vowels_count("abcde")
    2
    » vowels_count("ACEDY")
    3
    """
    if s == "": return 0
    cnt = len(list(filter(lambda ch: ch in "aeiouAEIOU", s)))
    if s[-1] in "yY": cnt += 1
    return cnt
```

This is a typical example where many-shot ICL benefits: with only a few input-output examples, the model might not be able to rule out the possibility of string length calculation, upper or lower case count, or judging whether there is a vowel (if all examples have only zero or one vowel). The special case of y is even trickier; the model can get a decent accuracy if it ignores y, but to achieve perfect reasoning, the model needs to find sufficient examples where y is at the end of the word and where y is not at the end of the word to eventually determine this special rule.

### E.4 Problem Types in Sec. 4.2

To study the effect on the topic of the problems for whether the problem benefits from many-shot, we first try to cluster the 693 problems in MIR-Extended using GPT-4o-0806. More specifically, we first prompt the LLM to generate python-style tags for each problem with the following prompt:

after acquiring tags for each problem, we prompt the LLM to merge all different tags down to 30
different tags with 6 major types: {List Analysis, List Manipulation, Mathematical Computations,
String Analaysis, String Manipulations, Other}. Tab. 13 shows the number of problems, detailed tags
and metric $D$ (defined in Eq. (1)) for each problem type.

Based on the results, we find that generally adding more problem will have a positive effect on
performance; however, for some types of problem such as geometric calculation and summation,
the performance will decrease with more shots included. Upon checking those problems, we found
them mostly fall into two categories: 1) the function is relatively straightforward, but the LLM gets
confused with more shots due to over-complicated guesses; 2) the function is too hard to guess, and
the LLM cannot make reasonable guesses when aggregating many pieces of information. The boxes
below give examples for case 1) and 2) respectively:

```
# Case 1: Straightforward Problems
[[Code]]
def solution(num: int) → int:
    steps = 0
    while num > 0:
        if num % 2 == 0: num /= 2
        else: num -= 1
        steps += 1
        return steps
[[Input-Output Pairs]]
Input: {'num': 68037}
Output: 23
. . .
```

```
# Case 2: Difficult Problems
[[Code]]
def solution(boardSize, initPosition, initDirection, k):
    yq, yr = divmod(initPosition[0] + k * initDirection[0], 2 * boardSize[0])
    xq, xr = divmod(initPosition[1] + k * initDirection[1], 2 * boardSize[1])
    # Calculate the final position considering reflections
    return [min(yr, 2 * boardSize[0] - yr - 1), min(xr, 2 * boardSize[1] - xr - 1)]
[[Input-Output Pairs]]
Input: {'boardSize': [10, 11], 'initPosition': [5, 9], 'initDirection': [1, -1], 'k': 264}
Output: [9, 9]
```

On the other hand, problems such as removing duplicates and string manipulation generally benefit
more from many-shot, probably because manipulation on each element / character can be considered
a shot by itself, and thus the effective number of "shots" in such types of problems are higher. Note,
"Others" problems have relatively high value of $D$ because of one outlier; other than the outlier, it is
almost equal to average level of $D$ across MIR-Extended.

## E.5   Other Factors Studied in Sec. 4.2

Fig. 11 illustrates the relation between our metric $D$ (see Eq. (1) for definition) for distinctiveness
between few-shot and many-shot performance.

Table 13: The tags for problem topics and related statistics; $D$ is the average metric (see Eq. (1) for definition) of the corresponding type of problems in MIR-Extended. We marked entries with $D > 0.1$ blue and $D < -0.1$ red. While increasing the number of shots generally brings better performance, We find that string manipulation benefits the most from many-shot.

| Major Tag | Minor Tag | # (MIR-Extended) | # (MIR-Core) | $D$ |
|---|---|---|---|---|
| List Analysis | Counting Elements | 4 | 2 | 0.075 |
| List Analysis | Counting Occurences | 25 | 9 | 0.017 |
| List Analysis | Maximum/Minimum Elements | 30 | 5 | -0.024 |
| List Analysis | Statistics | 2 | 0 | 0.041 |
| List Analysis | | 61 | 16 | 0.001 |
| List Manipulation | Filtering Elements | 30 | 14 | 0.066 |
| List Manipulation | Generating Sequences | 15 | 6 | 0.077 |
| List Manipulation | Mapping Elements | 9 | 3 | -0.022 |
| List Manipulation | Removing Duplicates | 6 | 4 | 0.136 |
| List Manipulation | Sorting Elements | 16 | 7 | -0.041 |
| List Manipulation | | 76 | 34 | 0.041 |
| Mathematical Computations | Basic Arithmetic | 35 | 12 | 0.085 |
| Mathematical Computations | Boolean Determination | 7 | 0 | 0.033 |
| Mathematical Computations | Calculations Based on Formulas | 98 | 32 | 0.051 |
| Mathematical Computations | Condition Checking | 43 | 20 | 0.114 |
| Mathematical Computations | Geometric Calculation | 4 | 2 | -0.110 |
| Mathematical Computations | Number Base Conversions | 12 | 5 | 0.038 |
| Mathematical Computations | Rounding | 7 | 5 | 0.185 |
| Mathematical Computations | Summation | 5 | 0 | -0.128 |
| Mathematical Computations | | 211 | 76 | 0.066 |
| String Analysis | Character Code Calculations | 14 | 3 | -0.004 |
| String Analysis | Comparison | 13 | 6 | 0.255 |
| String Analysis | Counting Characters | 29 | 10 | 0.050 |
| String Analysis | Pattern Matching | 27 | 7 | 0.084 |
| String Analysis | | 83 | 26 | 0.084 |
| String Manipulation | Case Transformation | 19 | 7 | 0.089 |
| String Manipulation | Encryption/Decryption | 8 | 4 | 0.061 |
| String Manipulation | Generating Substrings | 8 | 4 | -0.027 |
| String Manipulation | Rearranging Characters | 48 | 22 | 0.105 |
| String Manipulation | Substitution | 36 | 30 | 0.327 |
| String Manipulation | Substring Replacement | 33 | 22 | 0.205 |
| String Manipulation | Swapping Parts | 6 | 5 | -0.053 |
| String Manipulation | Transformation | 34 | 17 | 0.083 |
| String Manipulation | | 192 | 111 | 0.160 |
| Others | | 76 | 37 | 0.123 |
| Total | | 693 | 300 | 0.092 |

# F   More Experiment Details

## F.1   Prompts for Main Results

We provide the prompt for the main results in Sec. 4.1 and Sec. 4.2 in the box below (the first commented line is not a part of the prompt):

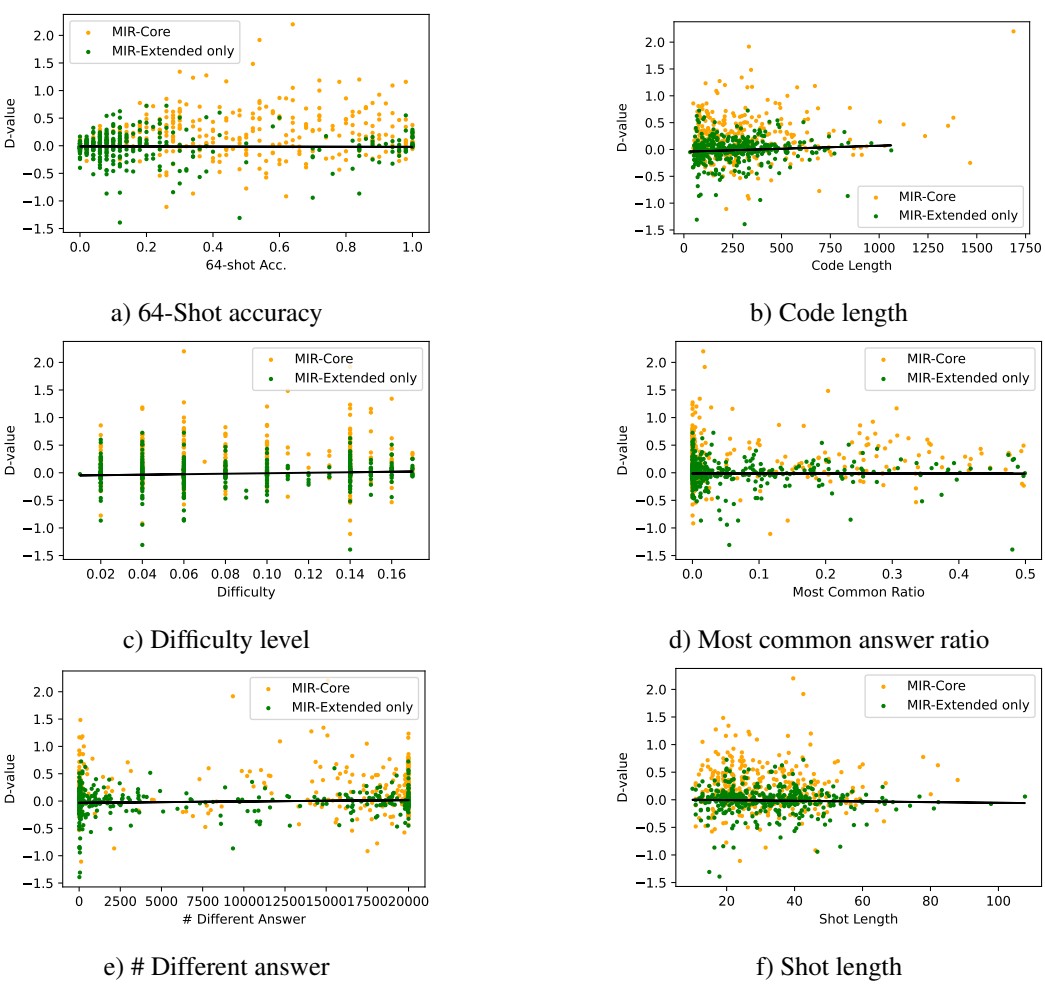

a) 64-Shot accuracy

b) Code length

c) Difficulty level

d) Most common answer ratio

e) # Different answer

f) Shot length

Figure 11: Single-factor analysis between each factor in Sec. 4.2 and our distinctiveness metric $D$. Each point represents one of the 693 functions in our benchmark. The black line is the linear regression result of all functions in MIR-Extended; it is clearly shown that $D$ is positively related to difficulty level and code length. The 64-shot accuracy is an average of {GPT-4o-0806, GPT-4o-mini-0718, Gemini-Pro 1.5-002, Gemini-Flash 1.5-002, Mistral-Large-2} over 10 test cases.

```
# prompt for main results

You are given some function that takes something as input and output something. You need to predict the
output for the target input of that function. Remember always end your answer with 'Output: your answer',
with your answer in strict python format. Here are some examples:
Input: <example input 1>
Output: <example output 1>
Input: <example input 2>
Output: <example output 2>
... (omitting more shots)
Input: <target input>
```

## F.2 Prompts for Ablations

**Effectiveness of CoT.** The following boxes demonstrate the prompt for the result used in Sec. 4.4 with forced CoT and no CoT respectively (the first commented line is not a part of the prompt):

# prompt for forced CoT

You are given some function that takes something as input and output something. You need to predict the output for the target input of that function. You need to first analyze it after 'Analysis:', then give your answer after 'Output:'. Remember always end your answer with 'Output: your answer', with your answer in strict python format. Here are some examples:
Input: <example input 1>
Output: <example output 1>
Input: <example input 2>
Output: <example output 2>
... (omitting more shots)
Input: <target input>

# prompt for no CoT

You are given some function that takes something as input and output something. You need to predict the output for the target input of that function. Your answer should always be 'Output: your answer', with your answer in strict python format. DO NOT OUTPUT ANYTHING ELSE INCLUDING YOUR THOUGHTS. Here are some examples:"
Input: <example input 1>
Output: <example output 1>
Input: <example input 2>
Output: <example output 2>
... (omitting more shots)
Input: <target input>

**Robustness of LLM intelligence.** The following box demonstrates the prompt for the result used in Sec. C.1. For the "unaware" setting, we use the same prompt as that in the main results; for the "aware error" and "aware ratio" setting, we use the following prompts respectively:

# prompt for "aware error"

You are given some function that takes something as input and output something. You need to predict the output for the target input of that function. Remember always end your answer with 'Output: your answer', with your answer in strict python format. Here are some examples. Note that not all shots are correct; there are a small portion of shots that are incorrect:
Input: <example input 1>
Output: <example output 1>
Input: <example input 2>
Output: <example output 2>
... (omitting more shots)
Again, note that not all shots are correct; there are a small portion of shots that are incorrect. Use your caution and think wisely.
Input: <target input>

# prompt for "aware ratio"

You are given some function that takes something as input and output something. You need to predict the output for the target input of that function. Remember always end your answer with 'Output: your answer', with your answer in strict python format. Here are some examples. Note that not all shots are correct; there are <number of error shots> out of <total number> shots that are incorrect:
Input: <example input 1>
Output: <example output 1>
Input: <example input 2>
Output: <example output 2>
... (omitting more shots)
Again, note that not all shots are correct; <number of error shots> out of <total number> shots that are incorrect. Use your caution and think wisely.
Input: <target input>

## F.3  Prompts for Reformatting APPS problems (Sec. 3.2)

The following box demonstrates the prompt for reformatting APPS problems in the "function collection" part of Sec. 3.2.

```
# prompt for reformatting
You are a coding expert. You will be given a problem and corresponding solution. Rewrite the solution
such that:
1. It becomes a single function named 'solution', which takes parameters as input instead of reading from
input() function if there is any;
2. There is no code out of the solution function and no solution class. All auxiliary functions should be
defined inside the solution function, and all imports should also be in the function.
3. The solution function should not have any print() function. Instead, it should return the result of the
function. If you need to output any rationale, leave them in comments. Your output must be directly
runnable without any change.
4. Just output the rewritten function; do not test it with extra statements.
Here is an example:
[[Problem]]
problem: Given a string, you need to reverse the order of characters in each word within a sentence while
still preserving whitespace and initial word order.
Example 1:
Input: "Let's take LeetCode contest"
Output: "s'teL ekat edoCteeL tsetnoc"
Note:
In the string, each word is separated by single space and there will not be any extra space in the string.
[[Solution]]
class Solution:
    def reverseWords(self, s):
        """
        :type s: str
        :rtype: str
        """
        rev_str = s[::-1]
        rev_arr = rev_str.split()
        final = rev_arr[::-1]
        return ' '.join(map(str, final))
[[Rewrite]]
def solution(s):
    """
    :type s: str
    :rtype: str
    """
    rev_str = s[::-1]
    rev_arr = rev_str.split()
    final = rev_arr[::-1]
    return ' '.join(map(str, final))
```

## F.4  Prompt for The Generation of Data Generator

The following box demonstrates the prompt for generating data generator:

```
# prompt for generating data generator
You are a coding expert. You will be provided a coding question and corresponding solution. Please write
two python function that randomly generates test case for the question. Specifically:
The first function's name is gen1, which generates random data (should be able to generate VERY
DIVERSE, i.e., at least 1000 different data points).
The second function's name is gen2, which generates data that is slightly harder than those generated in
gen1. (should be able to generate at least 100 different data points).
You shall not define any function outside gen1 or gen2. Should you use any helper function, make them
inner functions inside gen1 or gen2. You gen1 and gen2 function should have and only have one int
parameter, which is the number of cases.
Finally, the special cases should be designed as informative as possible that reveals the underlying function
when looking at the input and corresponding output from the solution.
Here is an example. Note the output of gen1 and gen2 should be a list of dicts describing the parameters,
and your special case input should be a dict describing the parameters. Please follow the format, and do
not generate cases that are too long. Do not output any other text; put all your thoughts after "# rationale:"
as shown in the example.
[[Problem]]
from typing import List
    def has_close_elements(numbers: List[float], threshold: float) -> bool:
    """ Check if in given list of numbers, are any two numbers closer to each other than given threshold.
    >>> has_close_elements([1.0, 2.0, 3.0], 0.5)
    False
    >>> has_close_elements([1.0, 2.8, 3.0, 4.0, 5.0, 2.0], 0.3)
    True
    """
from typing import List
[[Solution]]
sorted_numbers = sorted(numbers)
for i in range(len(sorted_numbers) - 1):
if sorted_numbers[i + 1] - sorted_numbers[i] < threshold:
    return True
return False
[[Gen1]]
# rationale: none
import random
def gen1(num_cases: int):
    low, high = 5, 10 # generate lists between length 5 to 10
    data = []
    for i in range(num_cases):
        N = random.randint(low, high)
        lst = [round(random.random() * 10, 1) for _ in range(N)]
        threshold = round(random.random(), 1) + 0.1
        data.append('numbers': lst, 'threshold': threshold)
    return data
[[Gen2]]
import random
def gen2(num_cases: int): # rationale: the data is slightly harder as the list is slightly longer
    low, high = 10, 20 # generate lists between length 10 to 20
    data = []
    for i in range(num_cases):
        N = random.randint(low, high)
    lst = [round(random.random() * 10, 1) for _ in range(N)]
    threshold = round(random.random(), 1) + 0.1
    data.append('numbers': lst, 'threshold': threshold)
    return data
```

## F.5 LLM Difficulty Labeling

In Sec. 4.2, we discussed several factors for problems which leads to LLM's few-shot and many-shot
performance difference, and one of them is LLM-evaluated function difficulty level. While seemingly
straightforward, a reliable evaluation of difficulty for LLM is highly non-trivial, as **LLM tend to
underestimate inductive reasoning difficulty when given a simple ground truth function.** We
provide an example below in the box with GPT-4o-latest as the prompted model:

# Prompt

You are an expert in coding and inductive reasoning. You will now be given a function that solves some problems in python and some example input-output pairs.

Please rate the difficulty of guessing the function through input-output pairs. You should judge the difficulty NOT on whether the function itself is easy to write, but on whether it is easy to guess. You should first give your rationale after [[Analysis]], then end your answer with 'Overall Score: [[x]]', where x is your score from 1 to 10.

[[code]]

```
def solution(l):
    return [l[i] for i in range(1, len(l)) if l[i] % i == 0]
```

[[data]]

Input: 'l': [40, -13]
Output: [-13]
Input: 'l': [-87, 29, -27, 90]
Output: [29, 90]
Input: 'l': [48, -4, 16, -63, -32, -5, -32, -45]
Output: [-4, 16, -63, -32, -5]
Input: 'l': [-75, 77, 44, 10, 17, -71, -80, -11, -68, -41]
Output: [77, 44]
Input: 'l': [87, 71]
Output: [71]

# LLM's response

[[Analysis]] To solve this puzzle, the key is to identify the rule used to filter the input list 'l'. By observing the given input-output pairs and the function definition, the pattern can be deduced as follows:

**Function Logic.** The function constructs a new list from the original list 'l', starting from index '1'. For each element at index 'i', it checks if the element is divisible by its **index** (using 'l[i] % i == 0'). If true, it includes the element in the resulting list.

**Difficulty Analysis Based on Input-Output Pairs.** **Clues that might help guess the function:** 1. **The output never includes the first element of the input list.** This suggests that index-based filtering is involved.

2. **Only specific elements from the input list appear in the output.** A guesser would likely try to find what distinguishes the selected elements from the others. Given the presence of both positive and negative numbers, divisibility by index is a plausible criterion.

3. **The divisibility condition isn't obvious at a glance.** It isn't immediately clear that elements are being checked for divisibility by their **index**, as it's somewhat unconventional. Most people would initially look for a rule involving the values themselves, like even/odd checks, thresholds, or signs.

**What Makes It Difficult to Guess?**

1. The divisibility rule by index is **non-intuitive**. Divisibility by **element values** is a common guess, but divisibility by **index** is not.

2. The function starts from index '1' instead of '0'. This subtle detail could mislead a guesser into thinking it's a filtering rule based on the **position** in the list rather than the index value.

3. The rule applies to **all elements**, regardless of their sign or magnitude. This broad applicability makes it harder to pin down the exact condition.

**What Makes It Easier to Guess?**

1. **Simple filtering condition:** Despite the challenge, once a guesser considers index-based filtering, the rule becomes easy to validate.

2. **The input-output pairs are consistent.** There are no exceptions or edge cases that would throw off a guesser.

**Overall Score: [[7]] (moderately difficult)**

GPT-4o-latest gives an average difficulty score of 7.2 / 10 out of 5 evaluations (which indicates moderate-to-high difficulty), while GPT-4o-0806 gives much lower difficulty score (below 5). This problem, however, is in fact so difficult that we have to exclude it from our benchmark, i.e., have 0 accuracy for all five models {GPT-4o-0806, GPT-4o-mini-0718, Gemini-Pro 1.5-002, Gemini-Flash 1.5-002, Mistral-Large-2} across {4, 8, 16, 32, 64, 128, 256, 512, 1024, 2048} shots in 10 test cases.

To address such issue, we adopt a multi-round conversation approach with self-reflection. We first let the LLM to try to solve the problem without code by itself, and then reveal the ground-truth answer and let LLM to score the difficulty based on self-reflection. The box below demonstrates an example:

With such paradigm, we found that the evaluation from LLMs are much more accurate.

## F.6 Answer Extraction

We use the following way to extract final answer:

1. find the last occurence of "utput:" in LLM's response, which can be either "output" or "Output";

2. remove trailing "**" after "output:" for bold, "'" and "'python in the answer for python format, and then remove all leading and trailing whitespaces and linebreaks;

3. conduct an exact match. If succeeds, the LLM is considered to be correct; otherwise, we try to convert the output into a python object by using ast.literal_eval(). If the convertion fails, the LLM is considered to be incorrect.

4. If the output can be converted to a single-key dictionary or single-element set, we will do an exact match between the value of the dictionary / element of the set to the ground truth answer with both converted to string (This is to account for responses similar to {"ans": 3} with ground truth being 3); otherwise, we do an exact match between the whole output and the ground truth answer converted to string.

## F.7 Robustness Test: Erroneous Shots

We generate test cases with erroneous shot in the following way:

1. For $n$-shot with a given error rate $ER$, randomly sample $ER \times n$ indices to be the "erroneous shots" with incorrect answer. $ER \times n$ is guaranteed to be an integer.

2. for each "erroneous shot", we randomly sample one unused shot as we generate 20000 shots for each function, and substitute the original output with the selected shot's output. We will re-sample the unused shot if its answer is identical with the original shot.

## F.8 SolverLearner

We use the following prompt for SolverLearner [10]:

```
# Prompt for SolverLearner
You are given some function that takes something as input and output something. You need to write a
python code of the function. You need to write your rationale after # (as if it is a python comment), and
give your answer after 'Code:'. DO NOT OUTPUT ANYTHING ELSE. Your function name should be
'solution'. You are not allowed to write other custom functions unless it is inside 'solution'. Use imports
before using package functions. You must strictly follow python format, especially input / output format
(e.g., if it is a dictionary, your param should also be a dictionary). DO NOT ADD ANY STATEMENT FOR
EVALUATION AFTER 'solution'. Here are the input-output pairs for the function, with input followed by
output:"
Input: <input 1>
Output: <output 1>
. . .
Input: <input n>
Output: <output n>
Here is your code. Again, do not output anything else; Your function name should be 'solution'. You are
not allowed to write other custom functions unless it is inside 'solution'. Use imports before using package
functions. You must strictly follow python format, especially input / output format (e.g., if it is a dictionary,
your param should also be a dictionary). DO NOT ADD ANY STATEMENT FOR EVALUATION AFTER
'solution'.
Code:
```

## F.9 Meta-Shots

We use the following prompt for the meta-shot experiments as illustrated in the box below:

You will be provided with a list of inductive reasoning problems, separated by '===================='.
Please answer the final problem as instructed by that problem, and refer to previous problems as examples.
====================
Input: <problem 1 input 1>
Output: <problem 1 output 1>
. . .
Input: <problem 1 input 8>
Output: <problem 1 output 8>
Input: <problem 1 target input>
<LLM CoT demonstration>
====================
Input: <problem 2 input 1>
Output: <problem 2 input 2>
. . .
Input: <problem 2 input 8>
Output: <problem 2 output 8>
Input: <problem 2 target input>
<LLM CoT demonstration>
====================
. . .
====================
Input: <target problem input 1>
Output: <target problem output 1>
. . .
Input: <target input>

