# OpenReview forum: "MIR-Bench: Can Your LLM Recognize Complicated Patterns via Many-Shot In-Context Reasoning?"
_NeurIPS.cc/2025/Datasets_and_Benchmarks_Track — NeurIPS 2025 Datasets and Benchmarks Track poster_

### Official Review · Reviewer_gzGj · 2025-07-02

**Rating:** 5
**Confidence:** 3

**Summary:**

This paper introduces MIR-Bench, the first benchmark designed to evaluate many-shot in-context reasoning (ICR) for pattern recognition in Large Language Models (LLMs). While existing benchmarks focus on few-shot (<10 examples) settings or simple classification tasks, MIR-Bench targets a more cognitively demanding capability: integrating hundreds to thousands of input-output examples to identify underlying functional patterns across diverse data formats—without requiring fine-tuning.

**Dataset Code Accessibility:**

Yes

**Ethical Considerations:**

No, there are no or only very minor ethics concerns

**Final Justification:**

The author rebuttal has addressed my concerns.

**Limitations Weaknesses:**

1.MIR-Bench tasks primarily focus on function approximation or input-output pattern learning, which represent a narrow subset of reasoning abilities.

2.Many-shot ICL with hundreds or thousands of examples requires models with large context windows (e.g., GPT-4-128K or Claude Opus).This may restrict the benchmark’s accessibility to only the most capable and expensive LLMs, limiting broad reproducibility and experimentation.

**Strengths Contributions:**

1.The paper is well-written and easy to follow.

2.The construction process of the MIR-Bench is clearly stated.

3.Compared with other datasets, this benchmark satisfies a broader set of characteristics.

4. Comprehensive analysis of factors affecting VLM performance are conducted, revealing that many-shot ICL is a promising but under-explored area.

---

> ### Author Rebuttal · Authors · 2025-07-28
>
> Thanks for your appreciation of our work. Here are our response:
>
> **Q1. Only testing narrow subset of reasoning abilities.**
>
> 1. While having diverse data sources such as math and logical reasoning problems can be beneficial for evaluating real-world ability, it may also introduce unexpected involvement of LLM's other abilities such as math. Instead, we intend to involve as little prior knowledge as possible so that the reason ability can be separately tested.
>
> 2. There are many related reasoning benchmarks which use only one form of data construction paradigm [1-3], including very influential ones such as ARC [1]. While real-world inductive/transductive problems can come from many sources, the essence of such reasoning is the model's ability to recognize and abstract of rules from examples. Compared to prior works, our work has multiple input-output modalities, which is a step forward in the diversity of reasoning ability evaluation.
>
> **Q2. Many-shot ICL requires large context windows, which may restrict the benchmark’s accessibility to only the most capable and expensive LLMs, limiting broad reproducibility and experimentation.**
>
> 1. The ability of dealing with long context is very important for today’s LLM community as it represents a growing demand where people except LLMs to complete more comprehensive and long-term tasks, such as completing a research project [4], beat an entire video game [5], or working in a large code repository [6]. Thus, there are many evaluation benchmarks proposed for LLMs [7, 8] which evaluates performance with large context windows; our dataset is also one such work.
>
> 2. There are already many open-source models that support long-context (128K-256K, an interval containing the context length of most models evaluated in the paper), such as Qwen 2.5, Qwen 3, Gemma 3 and Llama 3.3. Our proposed benchmark possesses an interesting challenge to these models.
>
> 3. While many-shot ICL requires more computational resources than supervised finetuning, it has two major advantages:
>
> a) many-shot ICL does not require any training, which is useful in model deployment and areas with only a small number of shots available (e.g. 100 or 200, which is too easy for supervised finetuning to overfit);
>
> b) models with supervised finetuning are often haunted with catastrophic forgetting issues [9], while many-shot ICL does not affect the model’s ability for dealing with a variety of tasks.
>
>
> **Reference**
>
> [1] F. Chollet et al. ARC Prize.
>
> [2] E. Banatt et al. WILT: A Multi-Turn, Memorization-Robust Inductive Logic Benchmark for LLMs. In ICLR, 2025.
>
> [3] Z. Yang et al. Language Models as Inductive Reasoners. In EACL, 2024.
>
> [4] S. Schmidgall et al. Agent Laboratory: Using LLM Agents as Research Assistants. arXiv: 2501.04227
>
> [5] Gemini Team. Gemini 2.5: Pushing the Frontier with Advanced Reasoning, Multimodality, Long Context, and Next Generation Agentic Capabilities. arXiv: 2507.06261
>
> [6] Q. Luo et al. RepoAgent: An LLM-Powered Open-Source Framework for Repository-level Code Documentation Generation. arXiv: 2402.16667.
>
>
> [7] C. Hsieh et al. RULER: What's the Real Context Size of Your Long-Context Language Models?. In COLM, 2024.
>
> [8] M. Wang et al. Leave No Document Behind: Benchmarking Long-Context LLMs with Extended Multi-Doc QA. In EMNLP, 2024.
>
> [9] S. Kotha et al. Understanding Catastrophic Forgetting in Language Models via Implicit Inference. In ICLR, 2024.

---

> > ### Comment · Reviewer_gzGj · 2025-08-05
> >
> > I appreciate the authors' response, which has addressed my concerns.

---

> > > ### Author Response · Authors · 2025-08-05
> > >
> > > Thanks a lot for your appreciation of our work and rebuttal! We will add related discussions to our revised version of paper to further clarify these concerns.

---

### Official Review · Reviewer_4sg1 · 2025-07-02

**Rating:** 5
**Confidence:** 4

**Summary:**

With models and their contexts both getting larger, there is a need to accurately evaluate the in-context reasoning abilities of the models. Authors highlight that existing in-context tasks / benchmarks either only test the few-shot ability of the models or are long-context, like Needle-In-A-Haystack, but do not test the ability to assimilate information across long spans. Moreover, few-shot reasoning does not allow for the expression of complex functions, such as when points are sampled from a higher-degree polynomial, which seems to require hundreds of in-context examples for a good representation. The authors propose MIR-Bench, which is a many-shot long-in-context reasoning benchmark. It is a pattern recognition benchmark requiring inducing a wide variety of underlying functions to solve it. Authors use their benchmark to evaluate current models and present studies across many paradigms, demonstrating that these models still lack long-context reasoning.

**Additional Feedback:**

L\<NUM\> denotes a reference to line number \<NUM\>.

Some typos and grammatical issues to be fixed:
- L96: "retrieval" instead of "retrival"
- L121: ...long before LLMs exist**ed**.
- L121: As LLMs approach~es~…

**Dataset Code Accessibility:**

Yes

**Dataset Code Comments:**

I checked the benchmark on Hugging Face, and looked at the code in the repository. The code appears to be well-documented and usable, thereby supporting reproducibility.

**Ethical Considerations:**

No, there are no or only very minor ethics concerns

**Final Justification:**

I think this is a well-written and timely paper that addresses an important gap in the benchmarking of language models. My evaluation is positive. I did not assign the highest score as I do think that the description of the benchmark in terms of types and examples of problems could have been done in a better way. I encourage the authors to incorporate that going forward.

**Limitations Weaknesses:**

- The benchmark is motivated as a long many-shot in-context reasoning benchmark; however, as presented by the authors in Table 1, there are existing benchmarks that test over longer contexts (many # shots). A proper comparison with these benchmarks is missing from the paper. Also, given the function, one can generate the desired number of I/O examples. Is the lack of examples due to difficulty in designing functions that would benefit from that many examples, or something else?
- There is a long line of work on understanding in-context learning (ICL) from the point of view of learning function classes. Some representative examples are [1], [2], and [3]. While they do not release any benchmark, the framework to evaluate ICL in these papers (introduced by [1]) is similar to that of MIR-Bench. I think that this should be covered under related work.
- In Figure 2, the performance eventually declines for all models after a certain number of examples. What do you attribute this drop to? Is it some fundamental lack of robustness in present-day models, or an issue in data generation? This is an interesting finding, but I would suggest that the authors ensure that the source is external to the proposed benchmark.
- While I understand the space constraints, I suggest the authors include some examples from the dataset in the main body of the paper to show the diversity and type of (inductive and transductive reasoning) problems being covered.


[1] Garg et al., What Can Transformers Learn In-Context? A Case Study of Simple Function Classes, NeurIPS 2022.

[2] Akyürek et al., What learning algorithm is in-context learning? Investigations with linear models, ICLR 2023.

[3] von Oswald et al., Transformers Learn In-Context by Gradient Descent, ICML 2023.

**Strengths Contributions:**

- I believe that the paper targets an important gap in LLM reasoning evaluation: we are investing in making the language models bigger and compatible with longer contexts, but we need to quantify reasoning over such contexts to conclusively determine the effectiveness of these investments. I liked the choice of inductive and transductive pattern recognition problems for this purpose and how they contrast with other benchmarks discussed in the paper (e.g., ARC).
- This work should have a large perceived impact as it inspires the field by highlighting a gap and advocates for focusing on such benchmarks. Moreover, the automated pipeline put forth by the authors to generate new tasks is a good step for continued development in this direction.
- The work is well-presented, easy to follow, and well-motivated.

---

> ### Author Rebuttal · Authors · 2025-07-28
>
> Thanks for your appreciation of our work. Here are our response:
>
> **Q1. There are existing benchmarks that test over longer shots, and one can generate desired number of I/O examples. Why other benchmarks lack examples?**
>
> 1. There are indeed papers that test more shots, which are manyICLbench [1] and HELMET [2]. The part of problems that involve >2048 shots in their paper, however, are all simple sentence classification problems such as TREC [3]; for example, the input is a short sentence “How did serfdom develop in and then leave Russia?”, and the output is “DESC” (description and abstract concept). Such kinds of tasks are largely limited in their input-output pattern.
>
> 2. The number of shots is limited by the context length of mainstream LLMs (which is 128K tokens) and thus cannot be infinite, as inputs that are too long will be meaningless with no model capable of dealing with them.
>
> 3. It is true that one can generate the desired number of I/O examples. This is why when we count the number of questions, we mentioned that **different sets of shots are not taken into account in the caption of Tab. 1; only the distinct target input-output pair is counted.** ManyICLBench [1], HELMET [2] and LongICLBench [4] are compilations of existing datasets, and thus their size is limited by the size of the underlying datasets. LMAct is a many-shot imitation learning benchmark with interactive environment (which makes it very hard to count the number of pattern recognition problems; we plan to change the “#PR prob.” in Tab. 1 and Tab. 3 to N/A with footnote in our revised version), and the scaling difficulty lies in finding suitable decision-making environments that are challenging but still solvable by LLM agents. The other many-shot works that exist in Tab. 3 but not in Tab. 2 are empirical studies instead of benchmarks, and thus they do not aim for scaling up data.
>
> **Q2. Missing papers in ICL evaluation.**
>
> Thanks for pointing these papers out! We will add a paragraph of discussion in our revised paper.
>
> **Q3. Why is the performance declining for all models after a certain number of examples?**
>
> 1. For 2048-shot, most models have significant performance decline because the input exceeds their maximum context length (see Tab. 8 for error rate jump from 1024 to 2048 shot; while some models such as GLM-4-Plus and Moonshot-128K has 0% error rate, they may conduct internal truncation).
>
> 2. For the reason of performance decline from saturation point (generally 128-256 shots) to 1024-shot, we present our hypothesis and validation in Sec. 4.3. In general, our hypothesis is that the saturation issue is not in information retrieval, but from distraction when aggregating too many pieces of information.  This is based on the following experiment result in Sec. 4.3:
>
> **a) many-shot vs. 16-shot duplicated evenly:** the performance gain from “having the same context length but more shots” first increases then decreases, which shows the problem is not context length but more pieces of information.
>
> **b) 16-shot duplicated evenly vs. 1-shot duplicated and 15-shot hided in it:** different difficulty for information retrieval with not much performance difference, which shows the problem is not information retrieval but aggregation.
>
> 3. We ensure that the drop of performance is external to the proposed benchmark by ensuring that the shots included in 2048-shot is a superset of 1024 and less shots (similar for other numbers of shots), as mentioned in line 273. This means, the information in more shots is monotonically increasing (except for the erroneous shot experiment), and thus for an ideal LLM, the performance will not drop.
>
> **Q4. Examples from the dataset and type of problems being covered in the main paper.**
>
> Thanks for the suggestion! Currently, we have examples of problems in Appendix E.2 and F.5, and a detailed problem topic analysis in Tab. 11. We will put a more concise version of these into the main body of our revised paper.
>
> **Q5. Typos in the paper.**
>
> Thanks for pointing these out! We will fix accordingly in our revised paper.
>
> **References**
>
> [1] K. Zou et al. On Many-Shot In-Context Learning for Long-Context Evaluation. In ACL, 2025.
>
> [2] H. Yen et al. HELMET: How to Evaluate Long-Context Language Models Effectively and Thoroughly. In ICLR, 2025.
>
> [3] Text REtrieval Conference (TREC) Question Classification dataset.
>
> [4] T. Li et al. Long-context LLMs Struggle with Long In-context Learning. arXiv: 2404.02060.

---

> > ### Comment · Reviewer_4sg1 · 2025-08-02
> > **Thanks for the rebuttal**
> >
> > I thank the authors for their rebuttal. My evaluation of this work remains positive, and I maintain my rating of 5.

---

> > > ### Author Response · Authors · 2025-08-02
> > >
> > > Thanks a lot for your appreciation of our work and timely reply! We will revise the paper according to the reviewer's advice.

---

### Official Review · Reviewer_4PtQ · 2025-07-03

**Rating:** 4
**Confidence:** 3

**Summary:**

This paper proposes a many-shot in-context reasoning benchmark for pattern recognition that asks LLM to predict output via input-output examples from underlying functions with diverse data format.

**Dataset Code Accessibility:**

Partly

**Ethical Considerations:**

No, there are no or only very minor ethics concerns

**Final Justification:**

Thanks for authors' rebuttal. I will raise the score because some concerns are addressed. I hope the author can included the responded contents into final draft.

**Limitations Weaknesses:**

Weakness:

1. While the authors claim MIR-Bench focuses on pattern recognition reasoning, it would be beneficial to provide clearer categorization or taxonomy of the reasoning tasks covered (e.g., which are mainly inductive vs. transductive, which require arithmetic reasoning vs. logical inference).

2. The paper provides an interesting quantitative analysis of factors influencing problem difficulty (Section 4.2), but lacks qualitative examples illustrating how these factors manifest in specific functions. Providing concrete examples of "difficult" vs. "easy" functions and explaining why many-shot context is required for each would greatly improve interpretability of the benchmark.

3. The paper would benefit from a clearer analysis of the dataset diversity (e.g., in terms of function types, input/output modalities, and complexity levels), a more detailed discussion of the potential downstream tasks that could leverage this benchmark, and a justification of the chosen evaluation metrics.

**Strengths Contributions:**

1. They acquired many insightful findings including scaling effect, robustness, inductive vs. transductive reasoning, retrieval Augmented Generation (RAG), coding for inductive reasoning, cross-domain generalizability, etc.

---

> ### Author Rebuttal · Authors · 2025-07-28
>
> Thanks for your constructive advice. Here are our response:
>
> **Q1. Taxonomy of the reasoning tasks covered.**
>
> We have a detailed list of tasks covered in Tab. 11 of our paper in Appendix E.3, with analysis on which type of tasks benefit the most from many-shot In-Context Learning (ICL).
>
> **Q2. Lack of qualitative examples showing how factors manifest in specific functions.**
>
> Thanks for pointing out; we have some discussion on difficulty and examples of problems with different difficulties in Appendix E.2 and F.5. We also provide an example of how many-shot helps in introduction line 38-42.
>
> Generally, according to our Fig. 11 c), the difficulty is positively correlated with the benefit of many-shot ICL. Here we append some more examples of difficult and easy questions in our benchmark and analysis on whether they could be benefited from many-shot ICL:
>
> Example of a very easy question:
>
> ```
> def add(x: int, y: int): # normalized difficulty level 0.1
>     """Add two numbers x and y
>     >>> add(2, 3)
>     5
>     >>> add(5, 7)
>     12
>     """
>     return x + y
> ```
>
> such a function is extremely easy to induce, and thus few-shot ICL is sufficient; LLMs will not benefit much from many-shot ICL.
>
> Example of a slightly harder question:
>
> ```
> def find_sum(arr):  # normalized difficulty level 0.4
>     return sum(set(arr))
> ```
>
> This question is slightly harder as it involves two operations: first remove all duplicate elements in a list, then get the sum. The model will need to look at multiple examples with and without duplicated elements to rule out other possible functions, e.g. sum of the array, fraction of the sum of array, etc. Thus, many-shot ICL will help more than the last function.
>
>
> Example of a more difficult question:
>
> ```
> def vowels_count(s): # normalized difficulty level 0.7
>     """Write a function vowels_count which takes a string representing
>     a word as input and returns the number of vowels in the string.
>     Vowels in this case are 'a', 'e', 'i', 'o', 'u'. Here, 'y' is also a
>     vowel, but only when it is at the end of the given word.
>
>     Example:
>     >>> vowels_count("abcde")
>     2
>     >>> vowels_count("ACEDY")
>     3
>     """
>
>     if s == "": return 0
>     cnt = len(list(filter(lambda ch: ch in "aeiouAEIOU", s)))
>     if s[-1] in "yY": cnt += 1
>     return cnt
> ```
>
> This is a typical example where many-shot ICL benefits: with only a few input-output examples, the model might not be able to rule out the possibility of string length calculation, upper or lower case count, or judging whether there is a vowel (if all examples have only zero or one vowel). The special case of y is even trickier; the model can get a decent accuracy if it ignores y, but to achieve perfect reasoning, the model needs to find sufficient examples where y is at the end of the word and where y is not at the end of the word to eventually determine this special rule.
>
>
> **Q3. Analysis on the dataset diversity.**
>
> 1. We have some analysis on the dataset diversity in the paper; for example, the taxonomy of problem topics is listed in Tab. 11, and (relative) difficulty level is plotted in Fig. 11 c); we will update the figure with actual values of difficulty level in the revised version of the paper.
>
> 2. To further address the reviewer’s concern, here we provide a list of problem difficulty in the following table, with normalized difficulty from 0 to 1 (higher is more difficult):
>
> Difficulty level | # MIR-Core | # MIR-Extended
> ---|---|---
> 0.05 | 0 | 1
> 0.1 | 21 | 81
> 0.2 | 53 | 103
> 0.3 | 50 | 118
> 0.35 | 1 | 1
> 0.4 | 27 | 51
> 0.45 | 0 | 2
> 0.5 | 33 | 75
> 0.55 | 5 | 10
> 0.6 | 2 | 7
> 0.65 | 4 | 8
> 0.7 | 77 | 163
> 0.75 | 8 | 25
> 0.8 | 13 | 32
> 0.85 | 6 | 16
>
> 3. Our benchmark includes a variety of questions with different input-output modalities. To demonstrate this, we list the input-output modality of the first 12 problems in MIR-Extended:
>
> str, str -> str
>
> str -> int
>
> list[int] -> list[int]
>
> str -> str
>
> list[int] -> list[int]
>
> list[int] -> list[int]
>
> int, int -> str
>
> int, int -> int
>
> str -> str
>
> str -> str
>
> int, int -> int
>
> int -> int
>
> which contains 7 different types of input-output modalities.
>
> We also list the input-output modality count in MIR-Core and MIR-Extended:
>
> **MIR-Core:**
>
> Modality | Count
> ---|---
> str -> str | 86
> int -> int | 18
> list[float] -> list[float] | 11
> int, int -> int | 9
> int -> str | 8
> str, str -> str | 8
> str -> int | 8
> list[int] -> int | 5
> str -> list[str] | 5
> list[str] -> list[str] | 4
> list[int] -> list[int] | 4
> float -> int | 3
> float -> float | 3
> list[float], list[float] -> list[float] | 3
> int, int, int -> int | 3
> int, int, int -> str | 3
> str, str -> int | 3
> str -> list[int] | 3
> list[float] -> float | 2
> int, int -> str | 2
> list[str] -> int | 2
> int -> list[int] | 2
> float -> str | 2
> list[float],  float -> list[float] | 2
> list[str] -> str | 2
> float, float -> float | 2
> list[float] -> int | 2
> str, int -> int | 2
> others | 93
>
> **MIR-Extended:**
>
> Modality | Count
> ---|---
> str -> str | 158
> str -> int | 40
> int -> int | 37
> list[float] -> float | 24
> int, int -> int | 24
> str, str -> str | 22
> list[int] -> int | 19
> list[float] -> list[float] | 18
> list[float] -> int | 17
> int -> str | 13
> str, int -> str | 10
> float, float -> float | 10
> list[int] -> list[int] | 10
> str -> list[str] | 8
> int ,int -> list[int] | 6
> float, float, float -> float | 6
> int, int -> str | 6
> str, str -> int | 5
> int, int, int -> int | 5
> float -> int | 5
> list[float], float -> list[float] | 5
> float -> str | 5
> others | 240
>
> which shows very high versatility in input-output modality.
>
> **Q4. More detailed discussion of the potential downstream tasks.**
>
> Thanks for pointing out; we will add the discussion in the revised version of paper. Specifically, here we provide two types of downstream tasks that are directly related to this benchmark - decision-making agents and Programming-by-Example (PbE).
>
> **1. Decision-making agents:** When making-decisions, online interaction can be costly and dangerous (e.g. controlling a robotic arm, or navigating Amazon and buy items). A good LLM agent should be able to learn from the past interactive experiences in the environment, either by building the back-ground dynamics model (which corresponds to model-based reinforcement learning) or imitating past expert behavior (which corresponds to imitation learning). Our benchmark fits into this type of application as the input can be seen as state-action pairs and the output can be seen as the outcome (utility gained and new states). See LMAct [1] for an example application.
>
> **2. Programming-by-Example (PbE):** PbE is a long-studied program synthesis paradigm where the LLM needs to write code based on input-output examples, which is widely used in coding assistant and Excel sheets autofill. While the main evaluation on our benchmark does not involve writing code, we do have evaluations on writing code for our input-output pairs in Appendix C.2; with minor modification, our benchmark can serve as a solid basis for the downstream PbE works, such as Wei et al. [2].
>
>
> **Q5. Justification of the chosen evaluation metrics.**
>
> Thanks for pointing out. Specifically:
>
> 1. The justification for evaluation metrics of “benefit from many-shot” (Eq. 1) is in line 222-226.
>
> 2. The justification for the selection of factors studied relevant to the problems’ distinctiveness between few-shot and many-shot is in line 227-233.
>
> 3. The justification for the designing choice of using quadratic function to fit the distinctiveness metric is in line 235-237.
>
> 4. The justification for using exact match is specified and explained in Appendix F.6, line 816-827. We choose exact match as the metric because the answers are unique and verifiable; therefore, there is no need to introduce extra possibility of incorrect judgement and computational cost by setting up a LLM judge.
>
> **Reference**
>
> [1] A. Ruoss et al. LMAct: A Benchmark for In-Context Imitation Learning with Long Multimodal Demonstrations. In ICML, 2025.
>
> [2] A. Wei et al. CodeARC: Benchmarking Reasoning Capabilities of LLM Agents for Inductive Program Synthesis. In COLM, 2025.

---

> ### Author Response · Authors · 2025-08-06
>
> Dear Reviewer 4PtQ,
>
> Thank you for your dedication into reviewing our paper and make it a better work. As the author-reviewer period is close to its end, we would like to know whether there is any unaddressed concern of the paper that we can help with. Have our rebuttal adequately addressed your concerns?
>
> Best,
> Authors

---

### Official Review · Reviewer_jUPA · 2025-07-18

**Rating:** 4
**Confidence:** 4

**Summary:**

This paper introduces a novel benchmark designed to evaluate a large language model's ability to perform complex pattern recognition through many-shot in-context reasoning.
MIR-Bench provides LLMs with hundreds to thousands of input-output examples generated by an unknown underlying function, tasking them with inferring the pattern to predict the output for a new input. Key findings from testing leading LLMs reveal that performance often saturates or even declines after a few hundred shots due to "attention dispersion," not a failure of information retrieval.
Overall, MIR-Bench provides a challenging new standard for assessing advanced reasoning in long-context LLMs and offers crucial insights into their current capabilities and limitations.

**Dataset Code Accessibility:**

Yes

**Ethical Considerations:**

No, there are no or only very minor ethics concerns

**Final Justification:**

The author's responses have resolved my queries. I support the acceptance of this paper.

**Limitations Weaknesses:**

The paper's central premise is to test a "primal ability for general intelligence" by asking LLMs to recognize "complicated patterns." However, the patterns are sourced exclusively from "introductory-level coding benchmarks"

To be honest, I'm not an expert in LLM evaluation.
From Table 1, the paper appears to address some subtle gaps in existing datasets, but the contribution doesn't strike me as particularly significant.
Given that the dataset is built by extending existing ones, I'm not convinced there's a pressing need for it.

Furthermore, Figure 5 suggests that the many-shot setting doesn't add more information for the models; the performance for most methods already saturates in the few-shot setting.
I don't think the task itself is unimportant, but rather that this specific dataset might not be providing genuinely new information with its additional examples. It appears that many of the samples may be redundant.

**Strengths Contributions:**

1. The work introduces a novel, well-justified, and highly significant benchmark that addresses a critical and previously unaddressed gap in LLM evaluation.
2. The paper provides a series of insightful and non-obvious empirical findings on the limitations of many-shot in-context reasoning, supported by rigorous ablation studies.

---

> ### Author Rebuttal · Authors · 2025-07-28
>
> Thanks for your constructive advice. Here are our response:
>
> **Q1. Source of problems.**
>
> 1. Introductory-level coding problems do not necessarily mean the underlying input-output pattern is easy to induce. For example, consider the problem *“given an input string and let ‘a’=1, ‘b=2’,...,’z’=26. If we see the string as the product of alphabets, output the last digit of the result; e.g., $f(bab)=(2\times1\times2)\ mod\ 10=4, f(zc)=(26\times3)\ mod\ 10 = 8$”*.This is very easy to implement but highly non-trivial to induce with only input-output pairs. A non-introductory level problem, such as dynamic programming problem with multiple functions and arrays of input, could be almost impossible to guess with input-output pairs even for humans. We find this in our early experiment and thus discard the more difficult coding questions. In fact, even with introductory-level problems, there are still a significant number of questions that are too hard to induce by LLMs or even humans. This is why in line 172-173, we need to “filter out problems that are unsolvable for all five LLMs in 10 test cases”.
>
> 2. Compared to many related and influential works that test LLM’s general reasoning ability, our benchmark takes a step forward in the diversity of input-output pairs. There are 7 different input-output modalities in the first 12 questions in MIR-Extended: str, str -> str; str -> int; list[int] -> list[int]; str -> str; int, int -> str; int, int -> int; int -> int. In contrast, many prior benchmarks, including very influential ones such as ARC Prize [1], only contain a single input-output modality (e.g. 2D array as input and output) [1-3]. (See reviewer 4PtQ Q3 for a detailed analysis on problem types.)
>
> 3. Having diverse data sources such as math and logical reasoning problems can be beneficial for evaluating real-world ability, but it may also introduce unexpected involvement of LLM's other abilities such as math. Instead, we intend to involve as little prior knowledge as possible so that the reason ability can be separately tested.
>
> **Q2. The contribution is not significant as it extends from existing datasets.**
>
> 1. Extending from existing datasets does not necessarily mean the dataset is not novel, as the existing dataset that we build on are for coding, i.e., the underlying **mapping rule is given**, but our dataset is about inductive/transductive reasoning, i.e. the underlying **mapping rule is unknown**; they are completely different tasks with different research scope (similar to rendering vs. 3D reconstruction). We select those tried-and-true functions as their input-output mapping is more practical in real-life applications.
>
> 2. Most prior many-shot In-Context Learning (ICL) benchmarks, such as manyICLbench [4], LongICLbench [5] and HELMET [6] fits more into this concern as they are mostly direct compilations of existing tasks, while we are not. **To overcome such limitations is our aim, which is the very meaning of the “new data” column in Tab. 1.**
>
>
> 3. The contribution of our work is threefold: a novel benchmark, an automatic pipeline and a complete, detailed empirical study with many insightful findings. The reviewer did not mention the latter two contributions. Also, with respect to benchmark, ours is the first many-shot benchmark that **is large-scale, has diverse input-output modalities, and overcomes “extending existing dataset limitation”**, and the first large-scale inductive/transductive reasoning benchmark that extends to many-shot settings.
>
> 4. In a high-level sense, the ability we test is a combination of LLM’s pattern recognition (inductive/transductive reasoning) and long-context ability. Both abilities are extremely important for today’s LLMs as the prior is a primal mental ability for intelligence and the latter represents a rapidly growing demand from the LLM community to let them handle complicated tasks, such as working on an entire repo [7] or conducting research [8]. Our work is a pioneering one in combining the two and filling in the gap from both worlds.
>
>
> **Q3. The dataset does not provide genuinely new information with many-shot as most models saturate in the few-shot setting.**
>
> 1.  We are not sure how Fig. 5 suggests that the model saturates in the few-shot setting. Fig. 5 is about studying the effect of CoT and whether directly outputting answer (transductive learning) outperforms CoT (inductive learning) for models, and is not the standard test setting of our paper. The results that show models stop to improve when more shots are given is in Fig. 2, where the saturation points are mostly between 64 and 256 shots. Such interval is generally considered as many-shot ICL, as normal ICL usually considers < 10 shot (see Tab. 1 “max # shots”).
>
> 2. It is true that for some problems that are too easy or too hard, new information is not necessary. This is exactly the reason why we conduct experiments in Sec. 4.2 and curate MIR-Core besides MIR-Extended: to select the problems that could benefit the most from many-shot ICL. The performance result in Fig. 2 show that many models have ~10% performance growth from 4-shot to many-shot, which is significant.
>
> 3. To further address the reviewer’s concern, we will add case analysis on our problems in the revised version to show how many-shot can help the problems with more complicated settings. Here is an example in our benchmark:
> ```
> def vowels_count(s): # normalized difficulty level 0.7
>     """Write a function vowels_count which takes a string representing
>     a word as input and returns the number of vowels in the string.
>     Vowels in this case are 'a', 'e', 'i', 'o', 'u'. Here, 'y' is also a
>     vowel, but only when it is at the end of the given word.
>
>     Example:
>     >>> vowels_count("abcde")
>     2
>     >>> vowels_count("ACEDY")
>     3
>     """
>
>     if s == "": return 0
>     cnt = len(list(filter(lambda ch: ch in "aeiouAEIOU", s)))
>     if s[-1] in "yY": cnt += 1
>     return cnt
> ```
>
> This is a typical example where many-shot ICL benefits: with only a few input-output examples, the model might not be able to rule out the possibility of string length calculation, upper or lower case count, or judging whether there is a vowel (if all examples have only zero or one vowel). The special case of y is even trickier; the model can get a decent accuracy if it ignores y, but to achieve perfect reasoning, the model needs to find sufficient examples where y is at the end of the word and where y is not at the end of the word to eventually determine this special rule.
>
> 4. It may be true that not every sample is useful, and there might be some redundant ones. However, this is also the case in real-life applications such as imitation learning [9] or programming by examples [10]. The many-shot ICL paradigm is designed to remove the need of carefully designing a pipeline to select "the most useful" examples by Retrieval-Augmented Generation (RAG) and utilize LLM's long-context ability and potentially less helpful but still usable data to the fullest.
>
> **References**
>
> [1] F. Chollet et al. ARC Prize.
>
> [2] E. Banatt et al. WILT: A Multi-Turn, Memorization-Robust Inductive Logic Benchmark for LLMs. In ICLR, 2025.
>
> [3] Z. Yang et al. Language Models as Inductive Reasoners. In EACL, 2024.
>
> [4] K. Zou et al. On Many-Shot In-Context Learning for Long-Context Evaluation. In ACL, 2025.
>
> [5] T. Li et al. Long-context LLMs Struggle with Long In-context Learning. arXiv: 2404.02060
>
> [6] H. Yen et al. HELMET: How to Evaluate Long-Context Language Models Effectively and Thoroughly. In ICLR, 2025.
>
> [7] S. Schmidgall et al. Agent Laboratory: Using LLM Agents as Research Assistants. arXiv: 2501.04227
>
> [8] Q. Luo et al. RepoAgent: An LLM-Powered Open-Source Framework for Repository-level Code Documentation Generation. arXiv: 2402.16667.
>
> [9] A. Ruoss et al. LMAct: A Benchmark for In-Context Imitation Learning with Long Multimodal Demonstrations. In ICML, 2025.
>
> [10]  A. Wei et al. CodeARC: Benchmarking Reasoning Capabilities of LLM Agents for Inductive Program Synthesis. In COLM, 2025.

---

> ### Author Response · Authors · 2025-08-06
>
> Dear Reviewer jUPA,
>
> Thank you for reading our rebuttal and updating your final score. As the author-reviewer period is close to its end, we would like to know whether there is any unaddressed concern of the paper that we can help with. Have our rebuttal adequately addressed your concerns?
>
> Best,
> Authors

---

> > ### Author Response · Authors · 2025-08-08
> >
> > Dear Reviewer jUPA,
> >
> > Thank you again for your time dedicated into reviewing our paper and reading our rebuttal. As the author-reviewer discussion period enters its last day, may we know that if our rebuttal has adequately addressed your concerns? We are happy to discuss any possible remaining issues if needed.
> >
> > Best,
> >
> > Authors

---

### Note · Authors · 2025-08-15

As the author-reviewer discussion period concludes, we would like extend our sincere gratitude to all reviewers, ACs and SACs for their constructive feedback, active engagement in discussion, and appreciation of our work and response.

We are delighted to see that the reviewers recognize our work as:

- novel, well-justified and highly significant **(Reviewers jUPA, 4sg1)**;

- well-presented, well-documented and easy to follow **(Reviewers 4sg1, gzGj)**;

- contains extensive, rigorous and insightful empirical findings **(Reviewers jUPA, 4PtQ, gzGj)**;

- with good dataset code accessibility **(Reviewers jUPA, 4sg1, gzGj)**;

- is constructive for continued development **(Reviewer 4sg1)**;

- and satisfies a broader set of characteristics compared to prior work **(Reviewer gzGj)**.

During the discussion, several questions about our paper were raised, including the dataset quality (justification of metrics by **Reviewer 4PtQ**, diversity, statistics and qualitative analysis by **Reviewers 4PtQ and 4sg1**, source of data by **Reviewers jUPA, gzGj)**, reproducibility with its long-context nature **(Reviewer gzGj),** clarity and contribution on comparison to prior works **(Reviewers jUPA, 4sg1),** potential downstream tasks **(Reviewer 4PtQ),** and further explanation on model saturation **(Reviewers jUPA, 4sg1).**

We have addressed these issues meticulously by providing statistics and case study, as well as clarification on topics including publicly available long-context LLMs, data source, comparison to prior work and model saturation issues. We are very pleased that all reviewers responded unanimously agree that all concerns have been resolved as no further concern is mentioned in reviewers' replies.

In summary, our benchmark is the **first many-shot reasoning benchmark that is large-scale, has diverse input-output modalities and is not a straightforward compilation of existing datasets**. We provide an **extensive, insightful and rigorous empirical study** that sheds light on the importance of many-shot in-context learning and provides many important insights, with an **available automatic pipeline that is constructive for future research in this area**.

Finally, we would like to thank all reviewers, ACs and SACs again for their efforts in strengthening our paper. We will integrate all the advice in the revision and are confident that it will be an important contribution to the community.

---

### Decision · Program_Chairs · 2025-09-18

**Decision:**

Accept (poster)

**Comment:**

This paper tests whether LLMs can recognize complex patterns, implemented by functions in code. An LLM is presented with input-output examples from a function and prompted to predict outputs for new inputs.

Strengths
1. Interesting new dataset for many-shot learning
2. Novel selection procedure for identifying examples that benefit from many shots

Weaknesses
1. Restricted to code functions. I don't think this is a fundamental limitation but I question whether these functions are all identifiable. That is, given N shots, is the function uniquely specified with low description length?
I would assume with high enough N, but I don't see this discussed in the paper.

Discussion:
1. Requests for more details and analysis are addressed

Overall I think the concerns are suitably addressed. I think the current literature on many-shot literature goes in a few different directions and this dataset establishes a different one than the existing work. That's a positive and a negative, as I think it breaks some new ground but doesn't necessarily offer clarity or provide a dataset that everyone in this area will use.  However, I can see this dataset being useful for its particular purpise.

One related paper the authors may wish to cite is:

Schwettman et al. "FIND: A Function Description Benchmark for Evaluating Interpretability Methods"

which has a similar flavor of querying certain function values and then inferring a function.